behaviour/evolution/structural biology

musculature, anatomy, locomotion, three-dimensional reconstruction, tomography

**Author for correspondence:**
Katrine Worsaae
e-mail: kworsaae@bio.ku.dk

# Muscular adaptations in swimming scale worms (Polynoidae, Annelida)

Marc C. Allentoft-Larsen[1], Brett C. Gonzalez[2], Joost Daniels[3], Kakani Katija[3], Karen Osborn[2,3] and Katrine Worsaae[1]

[1]Marine Biological Section, Department of Biology, University of Copenhagen, Universitetsparken 4, Copenhagen 2100-DK, Denmark
[2]Smithsonian National Museum of Natural History, Smithsonian Institution, P.O. Box 37012, Washington, DC 20013-7012, USA
[3]Monterey Bay Aquarium Research Institute, 7700 Sandholdt Road, Moss Landing, CA 95039, USA

MCA-L, 0000-0001-7978-1783; BCG, 0000-0001-6968-2677;
JD, 0000-0002-9480-6077; KK, 0000-0002-7249-0147;
KO, 0000-0002-4226-9257; KW, 0000-0003-0443-4298

Annelids are predominantly found along with the seafloor, but over time have colonized a vast diversity of habitats, such as the water column, where different modes of locomotion are necessary. Yet, little is known about their potential muscular adaptation to the continuous swimming behaviour required in the water column. The musculature and motility were examined for five scale worm species of Polynoidae (Aphroditiformia, Annelida) found in shallow waters, deep sea or caves and which exhibit crawling, occasional swimming or continuous swimming, respectively. Their parapodial musculature was reconstructed using microCT and computational three-dimensional analyses, and the muscular functions were interpreted from video recordings of their locomotion. Since most benthic scale worms are able to swim for short distances using body and parapodial muscle movements, suitable musculature for swimming is already present. Our results indicate that rather than rearrangements or addition of muscles, a shift to a pelagic lifestyle is mainly accompanied by structural loss of muscle bundles and density, as well as elongation of extrinsic dorsal and ventral parapodial muscles. Our study documents clear differences in locomotion and musculature among closely related annelids with different lifestyles as well as points to myoanatomical adaptations for accessing the water column.

# 1. Introduction

Annelida is a diverse animal group with about 21.000 extant species that have successfully colonized every marine environment, from benthic habitats in shallow waters to the deep-sea, and from the water column of the open ocean to that of caves [1–4]. In each environment, annelids exhibit different lifestyles and locomotion, which are mediated by muscles of the body wall, and of the parapodia if such are present [5–7]. The body wall musculature always includes longitudinal muscles and a variation of either circular or transverse muscles with additional bracing muscles. Intermediate diagonal and oblique muscles may support locomotion and extend into the parapodia [5,8,9]. Members of the large clade Errantia are generally motile and errant with well-developed parapodia, the muscles of which typically include acicular, chaetal, ventral, and dorsal muscles, as well as parapodial wall muscles [5]. However, the myoanatomy can vary greatly and there is no common muscular design for all annelids, nor within the two large and derived clades Sedentaria and Errantia [8,10–12].

Evolutionary flexibility in the muscular system in annelids has made it possible to adapt locomotion to the different needs of a wide variety of habitats, enabling further shifts to new ecological niches [13,14]. Swimming is frequently seen in annelids and is normally facilitated by a combination of body and parapodial movements in errantians. The longitudinal body wall muscles can by antagonistic contractions of left and right muscle groups create body undulations for propulsion. The movement of parapodia may add to the forward motion or even be the main driver of body movement [15]. The parapodial movements include power strokes (thrust gaining movement directed from head to tail) and recovery strokes (movements directed from tail to head towards starting position). Most benthic errantians may 'occasionally swim' [10,16–18], but are negatively buoyant and thus cannot passively maintain their position within the water column. Benthopelagic species are primarily associated with the sea floor but depend on entering the water column above it to feed or reproduce. Some benthic species have a pelagic reproductive stage (epitoky) that transforms the structure and sometimes the number of parapodia and chaetae, sensory organs, and musculature to swim into the water column to spawn [19–21]. Exclusively or continuously swimming errantian annelids have originated independently within multiple families and are here defined as animals capable of spending long periods of time suspended in the water column, maintaining their position there. These 'continuous swimmers' are found in the water column of the deep sea, open ocean and caves [22,23], with those occurring in the open ocean referred to as holopelagic. The transition to life in the water column opens up a different set of prey and a possible release from the high predation and competition of other benthic animals [18,24]. The transition to living exclusively in the water column has, in some cases, involved drastic morphological modifications such as transparent or gelatinous bodies and elongated appendages and parapodia [18,25,26]. The elongate parapodia are typically flexible with strong elongated extrinsic muscles and form elaborate oar-like structures (e.g. Tomopteridae; [27]). However, little is known about the evolution of specific morphological transformations facilitating their swimming lifestyle.

Morphological studies of benthic errantian annelids have suggested that short and less manoeuvrable parapodia with numerous muscle groups such as those found in Aphroditidae represent the plesiomorphic condition of recent Errantia [5]. This muscle arrangement may be useful when dragging an elongate body across various substrates or even burrowing into them [5,9]. Interestingly, the reduction of muscle groups, bundles, fibres and density may increase flexibility of the body and parapodia and has been suggested as an adaptation to life in the water column [5,11,28]. Additionally, studies of pelagic and cave species have suggested that elongated appendages may reduce sinking speed [11,22,28–30], in addition to expanding the volume of water they can sense predators and prey in.

Hence, we predict that swimming species compared to non-swimming relatives would possess parapodia with elongated but fewer, thinner and less dense muscle groups, thereby increasing buoyancy and flexibility at the same time as increasing velocity of the power and recovery stroke. However, even though the musculature of annelids has been studied since the early 1900s, these studies mainly focused on Aphrodite and Nereis (Nereididae) [16,29,31], with few studies describing in detail the myoanatomy of parapodia and their role in locomotion. New comparative studies of parapodial myoanatomy are therefore important for understanding parapodial evolution and functional significance for the diversification of lifestyles [5,8,9].

Scale worms (Aphroditiformia) are one of the most diverse annelid groups with more than 2000 described species [32]. Scale worms are characterized by the presence of scales (elytra) but exhibit highly divergent morphologies and lifestyles, inhabiting all known marine environments [3,32]. They

are largely epibenthic crawling predators with dorso-ventrally flattened bodies, well-defined parapodia and an eversible muscular pharynx with jaws [1,3], such as seen in the shallow water polynoid *Harmothoe imbricata* [10,33]. However, continued midwater oceanic exploration has revealed an increased number of continuously swimming scale worms, highlighting the group's success in diversification and colonization of new niches [1,17,22,25,34]. So far, only three truly holopelagic scale worm genera, i.e. *Drieschia, Drieschiopsis* and *Podarmus*, are described, and each is of unknown phylogenetic affinity [35–37]. The deep-sea group Macellicephalinae (Polynoidae) is generally considered epibenthic; however, two monotypic sister genera known exclusively from anchialine caves, *Gesiella jameensis* and *Pelagomacellicephala iliffei*, are found continuously swimming in the water column [38,39]. Occasional swimming or swimming bouts have also been observed for other deep-sea scale worms, including the commensal *Branchipolynoe* within the closely related Lepidonotopodinae (Polynoidae) [23]. These branchiate polynoids inhabit deep-sea mussels, swimming between mussel patches along with the bottom. While they cannot passively maintain their position in the water column, they are effective swimmers [17,40].

In order to establish hypotheses on morphological adaptations to swimming, we examined and compared the parapodial anatomy of five species of polynoid scale worms with different lifestyles. Their entire parapodial musculature were computationally three-dimensional reconstructed from micro-computed tomography (microCT) scans and the functionality predicted from comparison with locomotory recordings. Through interspecific comparisons, we wish to address whether the transition to continuous swimming involves changes in relative length, width, volume, groups, numbers and density of parapodial musculature.

# 2. Material and methods

## 2.1. Specimens and sampling

The five selected scale worms all belong to Polynoidae and span four different lifestyles and habitats: (i) benthic crawling, shallow water (*H. imbricata*, Polynoinae); (ii) continuous swimming, anchialine cave water column (*G. jameensis, P. iliffei*, Macellicephalinae); (iii) benthic crawling/possibly performing occasional swimming, deep-sea (*Macellicephala longipalpa*, Macellicephalinae) and (iv) occasional swimming, deep-sea commensal (*Branchipolynoe* sp., Lepidonotopodinae). Specimens were provided from personal scale worm collections of Brett C. Gonzalez, Marc C. Allentoft-Larsen, Katrine Worsaae, Paula Mendoza and from the Smithsonian National Museum of Natural History (table 1).

## 2.2. Micro-computed tomography

For later three-dimensional reconstruction, digitized three-dimensional images had to be created using microCT. All microCT scans were made at the Smithsonian National Museum of Natural History, Washington D.C., USA. Scans were obtained from at least two individuals from each species for a resolution of 102.0 µm voxel$^{-1}$ for *G. jameensis*; 112.6 µm voxel$^{-1}$ for *P. iliffei*; 71.208 µm voxel$^{-1}$ for *M. longipalpa*; 103.3 µm voxel$^{-1}$ for *H. imbricata* and 63.9 µm voxel$^{-1}$ for *Branchipolynoe* sp. Specimens not already fixed in ethanol were dehydrated through a dilution series to 70% ethanol. Postprocessing was done over a 2–3 step process from the initial fixative to DI-water, followed by additional 3–5 incremental steps from water to 70% ethanol. Once in 70% ethanol, all animals were placed in either individual vials or six-well plates covered with parafilm containing 0.3–0.6% PTA (phosphotungstic acid) in 70% ethanol. Samples were kept at room temperature with gentle rocking from 5 to 15 days. Fresh PTA was exchanged every 3–4 days. After staining, samples were washed with 70% ethanol over 2–3 days at room temperature with gentle rocking and regular rinsing. After rinsing, samples were prepared for scanning by being individually placed in approximately 5 mm diameter pipette tips filled with either 70% ethanol or with 0.5% low melt agarose made with DI-water. Tips were pre-sealed at the bottom by melting over a flame before adding the sample, ethanol or agarose. Melted paraffin was used to seal the tops of the pipette tips. To create a mount for the specimens, a single pipette tip was cut in half and hot-glued to approximately 2 mm carbon rod, then in turn, each specimen within their own pipette tip was nested within the halved pipette tip and secured using dental wax. All specimens were scanned using the nanotube (180 kV) on the GE Phoenix v | tome | x M 240/180 kV Dual Tube microCT machine.

**Table 1.** Information on examined species, sample localities, and numbers of specimens examined with each technique (μCT, VREC and CLSM). For *Pelagomacellicephala iliffei* and *Gesiella jameensis* habitat and sample specifics, see [41]. *Branchipolynoe* sp. was collected by manned submersible DSR Alvin. Specimens of *Harmothoe imbricata* were collected from small ropes overgrown with blue mussels (*Mytilus edulis*) hanging from a small pier in Kalbak, Faroe Islands and transported back in insulated containers to the laboratory at Marine Biology Section, University of Copenhagen and kept in an aquarium until video recordings and microscope examinations could be performed. Abbreviations: CLSM, confocal laser scanning microscopy; μCT, micro-computed tomography; SL, Specimen length; USNM, United States national museum number; VREC, video recording.

| species | sample locality | year | depth(m) | USNM | SL (mm) | μCT | VREC | CLSM |
|---|---|---|---|---|---|---|---|---|
| *Pelagaomacellicephala iliffei* | Grotto Hole, Long Island, Bahamas | 2007 | 1–2 | — | 9.4 | 2 | — | — |
| *Pelagomacellicephala iliffei* | Cottage Pond, North Caicos, Turks and Caicos | 2019 | 1.5 | — | 8.5 | — | 2 | 2 |
| *Gesiella jameensis* | Túnel de la Atlántida, Lanzarote, Canary Island, Spain | 2014 | 2–20 | — | 4.5 | 2 | — | — |
| *Gesiella jameensis* | Cueva de los Lagos, Lanzarote, Canary Islands, Spain | 2019 | 1.5 | — | 4.2 | — | 4 | 2 |
| *Macellicephala longipalpa* | 70° 51′ N. 52° 01′ W, West Greenland | 1928 | 733 | 51 968 | 17.3 | 1 | — | — |
| *Harmothoe imbricata* | Kaldbak, Kaldbak Fjord, Faroe Island | 2018 | — | — | 12.3 | 2 | 3 | 3 |
| *Branchipolynoe* sp. | 37° 40′ 21.0″ S 110° 52′ 37.2″ W, South Pacific | 2005 | +2236 | 1463 999 | 13.6 | 2 | — | — |

## 2.3. Computational three-dimensional reconstruction, volume and morphometric estimates

In order to analyse and compare muscle structures and muscle groups, computational three-dimensional reconstructions were made. MicroCT scanning files were imported individually as image stacks to the visualization software AMIRA 6.2.0 (ThermoFisher Scientific, Waltham, MA, USA) in order to create three-dimensional reconstructions of the parapodial muscle complex for each species. Only one well-imaged, scale-bearing, middle parapodium was chosen for reconstruction in each species, which has been shown to be sufficient for characterization of annelid parapodial features [42]. The three-dimensional reconstructions were carried out by manually identifying and demarking the boundaries of each parapodial muscle bundle from the microCT scan stacks in AMIRA. Muscles were generally demarcated in every 10th image of a stack, or in smaller intervals when muscle structures were very detailed or branched. Complete three-dimensional reconstructions of all muscles for each specimen were computationally extrapolated and smoothed from the marked images of the stack.

Note: An incomplete reconstruction of the holopelagic scale worm *Drieschia* sp. was made, but not included due to damaged tissue and deformities. It is therefore not included in the examined species. It is, however, worth mentioning that the overall musculature of *Drieschia* sp. seems to be less developed in terms of muscle bundles, mass and complexity than even *G. jameensis* and *P. iliffei*.

Volume estimates in $mm^3$ for each marked muscle in one specimen of each of the investigated species were calculated using the integrated measure tool in AMIRA. Morphometric measurements were obtained from two specimens of each species using an integrated measuring tool in AMIRA. The following procedures were followed: Parapodial length was measured on five middlemost segments from the lateral body wall to the distalmost tip of the neuropodium. The only exception to this was for *M. longipalpa* in which parapodia from segments 5–9 were used due to lack of suitable parapodia. The widths of the respective segments were measured as the transverse distance between the lateral-most point of the two ventral longitudinal muscles. Parapodial length was divided by the respective segment width to get the relative parapodium length. A dimensionless size correlated measure of volume was obtained by dividing muscle volumes by respective segment volume and illustrated in tables as (corr.). To use a mean parapodia length for each species, a *t*-test was performed using Excel 2016 to estimate significant differences between parapodial length of the two specimens of each species. An ANOVA test was then performed to see if the mean parapodial length of each species were significantly different from each other. Only one specimen could be obtained for *M. longipalpa* and therefore no *t*-test could be performed.

## 2.4. Confocal laser scanning microscopy

Confocal laser scanning microscopy was not used to reconstruct muscles but only to verify complex muscle structures. Specimens of *H. imbricata* and two specimens of *G. jameensis* and *P. iliffei* were cut into fragments and fixed overnight (O/N) in 4% paraformaldehyde in 0.15M PBS with 5% sucrose, then rinsed in PBS and stored with 0.05% NaN$_3$. Prior to staining, specimens were preincubated for 2 h in PTA buffer (0.15 M PBS + 1% TritonX + 0.25% BSA + 0.05% NaN$_3$ + 5% sucrose), thereafter incubated O/N at room temperature on a rocking table in 0.33 µM Alexa flour 633 Phalloidin (Invitrogen, ThermoFisher Scientific) diluted in PTA (with 0.1% TritonX). After rinsing in PBS, specimens were mounted directly between object glass and cover slide in fluoromount G (W/Dapi) (ThermoFisher Scientific) and held at 4°C refrigerator for 1–2 days and afterwards stored at −20°C until examination. Specimens were scanned using an Olympus Fluoview FV-1000 CLSM (Worsaae Lab, University of Copenhagen, Denmark). Imaris 7.7 (Bitplane Scientific Software) was used to analyse and create maximum intensity projections of Z-stacks.

## 2.5. Video analyses

To get a better understanding of the annelid locomotion and comparison between species, video recordings were carried out. All video recordings were taken within 24 h from species sampling. High-speed recordings of *P. iliffei*'s locomotion were taken using a monochrome camera (Sanstreak Edgertronic SC1) with different lens configurations: 2 x microscope Olympus objective with adapter and extension tubes and Nikon 50 mm f/1.8 lens with 12 mm or 20 mm extension tubes) at 900 fps. Extension tubes were used to achieve proper focus with the microscope objectives and to reduce the minimum focus distance with the other objectives. Brightfield illumination was obtained by use of LED light sources and a large-diameter (200 mm) condenser lens. Locomotion of *G. jameensis* and

*H. imbricata* were recorded using a Panasonic HC-VX980 camcorder respectively at 50 and 25 fps. Locomotion recordings did not include *M. longipalpa* nor *Branchipolynoe* sp. due to lack of live specimens. *Pelagomacellicephala iliffei* was recorded in a square plexiglass water tank 18 × 15 cm, whereas *H. imbricata* was recorded in a 250 ml plastic algae culturing vessel. *Gesiella jameensis* was recorded in a 9 cm Petri dish with black velvet background. All containers were filled with their respective ambient cave or sea water and had adequate space for free swimming. Selected sequences were digitized and analysed using the software DLTdv8 [43]. Locomotion analyses included visual observations of movement patterns of the body and parapodia and overall locomotion behaviour such as crawling or swimming. The parapodial movements were characterized in both crawling and swimming locomotion and described in steps from 1 to 5 (five steps for crawling). Parapodial angles were estimated directly from video frames using the DLTdv tool. Parapodial velocity was likewise estimated directly from video frames, but no quantitative data could support the analysis due to the inconsistency in the recording quality.

# 3. Results

Body movement in scale worms is generated by both body wall and parapodial musculature. The longitudinal body muscles are necessary for the body undulations to take place and comprise two paired bands of muscles (ventral and dorsal) that extend through the entire animal without segment separation. However, since the longitudinal muscles do not seemingly vary between the examined species, we have focused on comparing the parapodial muscle structures. The parapodial musculature of the five studied polynoids comprises 20–21 muscle groups with 51–67 individual sub-bundles, listed with keywords in table 2 and largely following the nomenclature of [8,42] and [10] (tables 2 and 3; electronic supplementary material, figure S1; figures 1–6). Main characteristics of each muscle and species are given below; for detailed descriptions of individual muscle groups in each species see electronic supplementary material (muscle group descriptions).

## 3.1. Parapodial muscles

In all species examined, the extrinsic dorsal muscles #I and II (presumed levator, #I, II, remotor, #II and promotor, #I muscles) and ventral muscles #III and IV (presumed remotor, #IV, promotor, #III and depressor, #III, IV muscles) constitute the most massive bundles, but numerous minor muscle groups are located intrinsically in the parapodium. The dorsal and ventral muscle groups #I–IV comprise complimentary anterior-posterior bundles, of which the anterior bundles are wider and thicker proximally (figures 2 and 3). The elytrophore is supported by a minor intrinsic, ventro-dorsally oriented parapodial muscle bundle #XX (depressor muscle) sheathed by supportive muscles (figure 1, #XX). The notopodium is slender and supplied by fewer muscle fibres than the neuropodium. It completely lacks extrinsic muscles and only has minor branches of fibres from the intrinsic chaetal muscles #XIII, #XIV, #XVI and muscles #I and #III (figure 2; electronic supplementary material, figures S2–S5). The neuropodium is longer and broader and has both the extrinsic dorsal and ventral muscle groups (#I–IV) and the intrinsic muscles #XVII, #XVIII and #XIX (figure 1). The bracing muscles (#VII–#VIII) support the body wall between the parapodia (figures 2 and 4). Oblique muscles (#V–VI) extend from the ventral body midline to the body wall, internal to the ventral muscles (#III-IV), likely supporting parapodial movements (figure 1). Inside the parapodium are numerous intrinsic muscle groups, a ventro-dorsal crossing muscle (#IX), numerous anterior-posterior transverse muscles, likely for keeping hydrostatic pressure (#X), and chaetal and acicular protraction and retraction muscles to move the chaetae and aciculae (#XII–XIX) (figures 1 and 2; electronic supplementary material, figures S2–S5, #IX, X, XII–XIX). No obvious circular muscles of the body wall were observed in any species. A very delicate outer net of thin, crisscrossing muscles surrounding the parapodial walls was observed by CLSM and phalloidin staining, possibly representing the parapodial wall musculature of [42] (electronic supplementary material, figure S6).

The swimming and closely related cave species, *Gesiella jameensis* and *Pelagomacellicephala iliffei*, both have relatively long parapodia and fewer number of individual muscle bundles (table 3), which are generally relatively thin and elongated (table 2, figures 2–5; electronic supplementary material, figures S1 and S2). Interacicular muscles #XXI are lacking. Specifically, the dorsal and ventral muscles (#I–IV) comprised fewer slender muscle bundles compared to *H. imbricata*, *M. longipalpa* and *Branchipolynoe* sp.; however, their acicular and chaetal muscles (#XII–XIX) are more numerous (figures 2–5; electronic

**7**

**Table 2.** Muscle summary. Characteristics of muscles #I–#XXI are listed for each species, including numbers of bundles, size, density and shape. Italics text traits possibly related to elaborate swimming skills (see discussion section). Characteristics marked with asterisk may be difficult to recognize on figures; see detailed description in electronic supplementary material. Note: The ventral muscles (#III and IV) comprise the previously described obliquus ventralis' [9], later changed to 'parapodial oblique' muscle by [42] and the 'ventral parapodial muscles', which are called 'lattice structures' in the neuropodium [42].

| muscle group number (#) | muscle group | Gesiella jameensis | Pelagomacellicephala iliffei | Macellicephala longipalpa | Harmothoe imbricata | Branchipolynoe sp. |
|---|---|---|---|---|---|---|
| #I | anterior dorsal muscle | 1 bundle/long/thin | 1 bundle/long/thin | 1 bundle/long/thin | 4 bundles*/medium length/thick (sheet structure) | 1 bundle/long/thin (elongated branches) |
| #II | posterior dorsal muscle | 1 bundle/long/thin | 1 bundle/long/thin | 3 bundles*/long/thin | 3 bundles*/medium length/thick (sheet structure and prominent around elytrophore) | 1 bundle/medium length/thick (prominent around elytrophore) |
| #III | anterior ventral muscle | 1 bundle/thin | 1 bundle/thin | 1 bundle/thin | 2 bundles*/thick (sheet structure and distally branched) | 1 bundle/thin (sheet structure) |
| #IV | posterior ventral muscle | 1 bundle/long/thin | 1 bundle/long/thin | 1 bundle/long/thin | 1 bundle/medium length/thick/(sheet structure and distally branched) | 1 bundle/medium length/thick (distally branched) |
| #V | anterior oblique muscle | 1 bundle/long/thin | 3 bundles/long/thin | 1 bundle/long/thin | 4 bundles*/short/thick (distally merged) | 4 bundles/medium length thick (distally merged) |
| #VI | posterior oblique muscle | 1 bundle/long/thin | 3 bundles/long/thin | 1 bundle/long/thin | 4 bundles*/short/thick/(distally merged) | 3 bundles/medium length/thick (distally merged) |
| #VII | preceding and succeeding bracing muscles | 2 bundles*/long/medium thick/W-shape | 2 bundles*/long/medium thick/W-shaped | 2 bundles*/long/ thick/W-shaped | 3 bundles*/short/thin/W-shaped | 2 bundles*/long/massive/W-shaped |
| #VIII | additional bracing muscle | 1 bundle/spatula-shaped | 1 bundle/spatula-shaped | 1 bundle/spatula-shaped | 1 bundle/twisted | missing |
| #IX | dorso-ventral crossing muscle | 1 bundle/thin/parapodium wall insertion | 1 bundle/thin/dorsal longitudinal muscle insertion | 1 bundle/thin/parapodium wall insertion | 1 bundle/thick/dorsal longitudinal muscle insertion | 1 bundle/thin/dorsal longitudinal muscle insertion |
| #X | anterior-posterior transverse muscles | 6 bundles/medium length/thin/loosely packed | 6 bundles/medium length/thin/loosely packed | 6 bundles/short/medium thick/tightly packed | 6 bundles/medium length/thick/tightly packed | 11 bundles/medium length/thin/loosely packed in groups |

**Table 2.** (Continued.)

| muscle group number (#) | muscle group | Gesiella jamensis | Pelagomacellicephala iliffei | Macellicephala longipalpa | Harmothoe imbricata | Branchipolynoe sp. |
|---|---|---|---|---|---|---|
| #XI | ventral diagonal muscles | 2 bundles/medium length/thin | 2 bundles/medium length/thin | 2 bundles/long/thin | 4 bundles/medium length/thick | 2 bundles/long/thick |
| #XII | notopodial acicula muscles | 9 bundles/long/thin | 9 bundles/medium length/thin | 7 bundles/short/thin | 7 bundles/short/thick | 10 bundles/long/thin |
| #XIII | notopodial chaetal protractor muscles | 2 bundles | 1 bundle | 1 bundle | 2 bundles | 2 bundles (string structure) |
| #XIV | notopodial chaetal retractor muscles | 1 bundle/medium thick | 1 bundle/massive | 1 bundle/medium thick | 1 bundle/thick | 1 bundle/medium thick |
| #XV | notopodial chaetal sac muscles | 1 bundle/medium length/half-moon structure | 1 bundle/medium length/half-moon structure | 1 bundle/long/half-moon structure | 1 bundle/medium length/half-moon structure | 1 bundle/long/ellipse structure |
| #XVI | neuropodial acicula muscles | 10 bundles/long/thin | 10 bundles/long/thin | 8 bundles/short/thin | 8 bundles/short/thick | 7 bundles/long/thin |
| #XVII | neuropodial chaetal protractors muscles | 2 bundles (+2 suspensor muscles)/medium thick | 2 bundles/medium thick | 2 bundles/medium thick | 2 bundles/thick | 2 bundles/medium thick |
| #XVIII | neuropodial chaetal retractor muscles | 1 bundle/medium thick | 1 bundle/medium thick | 1 bundle/medium thick | 1 bundle/thick | 1 bundle/medium thick |
| #XIX | neuropodial chaetal sac muscles | 1 bundle/medium thick/half-moon structure | 1 bundle/medium thick/half-moon structure | 1 bundle/thick/half-moon structure | 1 bundle/medium thick/half-moon structure | 1 bundle/medium thick/half-moon structure |
| #XX | elytrophore muscles | 2 bundles/thin | 2 bundles/medium thick | 5 bundles/medium thick | 6 bundles/thick | 4 bundles/medium thick |
| #XXI | interacicular muscle | missing | missing | missing | 1 bundle/short/thick | 1 bundle/short/thin |

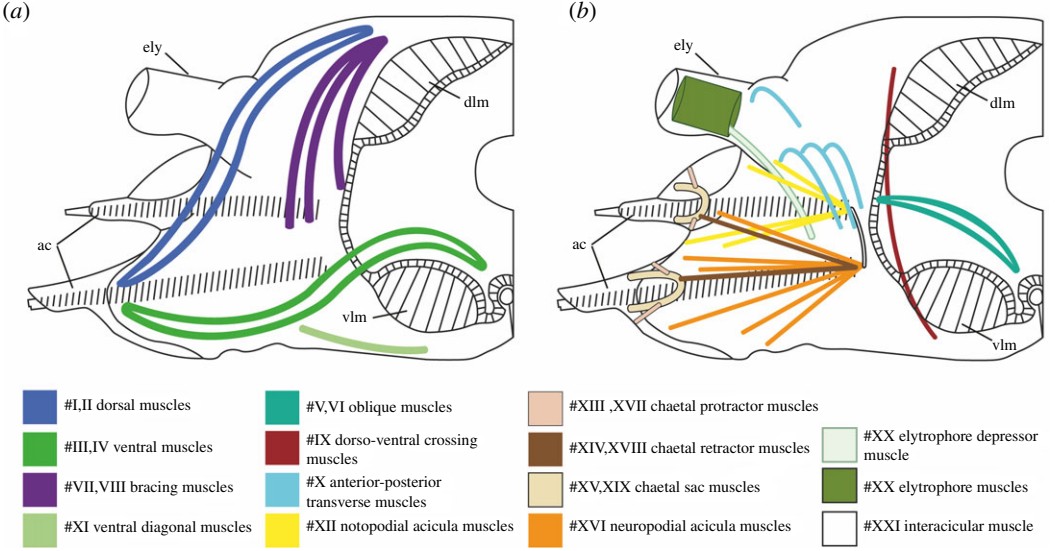

**Figure 1.** Schematic representation of the examined parapodial myoanatomy. Inspired by the results of *Harmothoe imbricata* [44]. All muscle groups are colour coded and numbered according to the legends. Dorsal is up in both drawings. (*a*) External parapodial myoanatomy. (*b*) Internal parapodial myoanatomy. Posterior bundles for simplicity are not shaded darker in this schematic illustration. Abbreviations; ac, acicula; dlm, dorsal longitudinal muscle; vlm, ventral longitudinal muscle.

supplementary material, figures S1 and S2). The notopodial chaetal sac musculature (#XV) was difficult to trace in *P. iliffei*, and only one obvious notopodial chaetal protractor muscle #XIII was found (versus two in *G. jameensis*; figure 5; electronic supplementary material, figure S2). *Pelagomacellicephala iliffei* also possesses three anterior and posterior oblique muscle bundles #V–VI, while *G. jameensis* only possesses one (table 2, figure 3; electronic supplementary material, figure S1). The supportive muscle in the elytrophore (#XX, figure 5; electronic supplementary material, figures S1 and S2) is more prominent in *P. iliffei* than *G. jameensis*.

The putatively swimming deep-sea species, *Macellicephala longipalpa,* belong to the same subfamily as the cave species and shows similarly long relative length of parapodia and similar muscle number and elongation, except for minor differences: (i) the parapodium is generally more slender, (ii) the notopodium is relatively smaller, (iii) bracing muscles are more prominent, (iv) acicular muscles are fewer, (v) the intrinsic muscles (#IX,X, XII–XIX) are generally more tightly packed (figures 4–6; electronic supplementary material, figure S3), (vi) the elytrophore muscles are more numerous and more prominent and (vii) the elytrophore is located more distally on the parapodium (figures 2 and 5; electronic supplementary material, figure S3) than seen in both *G. jameensis* and *P. iliffei*. For these comparisons, the age of the specimens should be considered because after 92 years in preservative, delicate tissue and muscles may have collapsed in the parapodium cavity, making the interior intrinsic musculature then also seem more tightly packed and shorter.

The benthic crawling species, *Harmothoe imbricata*, has a shorter relative parapodium length and additional individual muscle bundles; interacicular muscle #XXI is present (table 2 and figure 6). The muscle groups are generally relatively short, dense, thick, tightly packed and consist of numerous muscle bundles; the acicular and chaetal muscles are particularly dominant (table 2, figures 3–6; electronic supplementary material, figure S4). Ventral and dorsal muscles (#I–IV) comprise more muscle bundles and sheet-like structures than found in the other species (figures 2 and 3; electronic supplementary material, figure S3). The numerous intrinsic muscle groups (#IX, X, XII–XIX, figures 4 and 5) are tightly packed in the parapodium. The neuropodium is twice the size of the notopodium with the ventral and dorsal muscle groups (#I–IV) extending extrinsically. The intrinsic notopodial musculature #XIII–XV is likewise densely packed, but structurally follows that of the other species in number and thickness (figure 5). The elytrophore muscle #XX takes up a large portion of the parapodium with thick and dense muscle bundles extending both intrinsically and extrinsically (figure 5).

The occasionally swimming but benthic species, *Branchipolynoe* sp., has an intermediate relative parapodium length and an intermediate number of individual muscle bundles, which are generally relatively short and dense (table 2, figures 3–6; electronic supplementary material, figure S5). The muscular composition of *Branchipolynoe* sp. is most similar to *H. imbricata* when comparing muscle

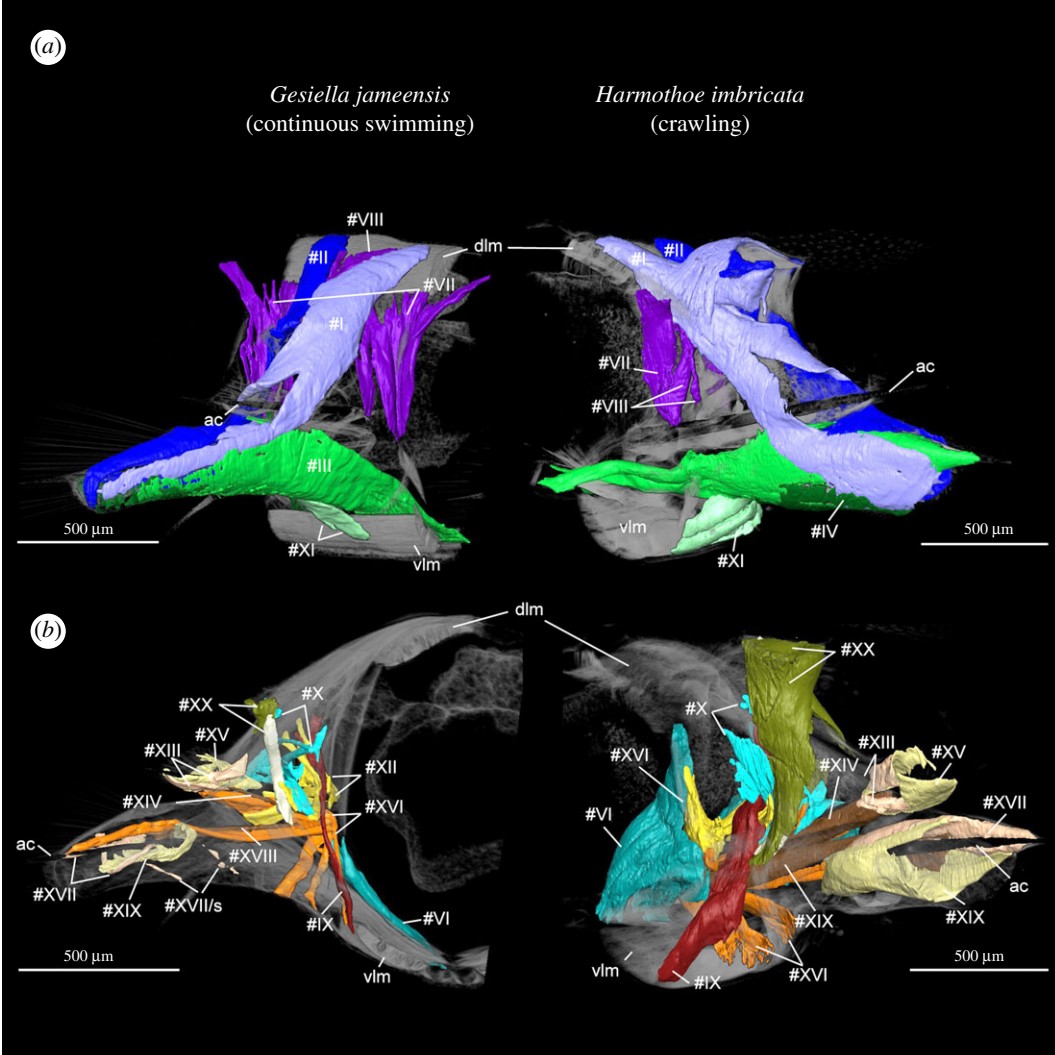

**Figure 2.** Comparison between external and internal myoanatomy of the middle segment, right-side parapodium in continuous swimming *Gesiella jameensis* and crawling *Harmothoe imbricata*. (*a*) External parapodial myoanatomy of *Gesiella jameensis* and *Harmothoe imbricata*, anterior view. (*b*) Internal parapodial myoanatomy of *Gesiella jameensis* and *Harmothoe imbricata*, posterior view. Dorsal is up in all images. Body surface is semi-transparent. Colour coding and numbering; #I, anterior dorsal muscle, blue; #II, posterior dorsal muscle, dark blue; #III, anterior ventral muscle, green; #IV, posterior ventral muscle, dark green; #V and VI, oblique muscles, blue-green; #VII, preceding and succeeding bracing muscles, purple; #VII, additional bracing muscles, purple; #IX, dorso-ventral crossing muscle, dark red; #X, anterior-posterior transverse muscles, turquoise; #XI, ventral diagonal muscles, lime green; #XII, notopodial acicula muscles, yellow; #XIII, notopodial protractor muscles, pink; #XIV, notopodial retractor muscle, brown; #XV, notopodial chaetal sac muscle, beige; #XVI, neuropodial acicula muscles, orange; #XVII, neuropodial protractor muscles, pink; #XVIII, notopodial retractor muscle, brown; #XIX, neuropodial chaetal sac muscle, beige; #XX, elytrophore muscles, olive green, (depressor muscle, egg white coloured); #XXI, interacicular muscle, white. Abbreviations; ac, acicula; dlm, dorsal longitudinal muscle; s, suspensor muscles; vlm, ventral longitudinal muscle.

bundle numbers and density. The most noticeable differences from *H. imbricata* are the large separations between the branches of the extrinsic muscles #I–IV, where #IV distinguishes itself by distinct branching (figure 3). The relative length of neuropodium to notopodium is shorter than in the other species examined, and the notopodium is shifted anteriorly relative to the neuropodium (electronic supplementary material, figure S5). Anterior-posterior transverse muscles are more numerous (#X, figure 4). A large cavity between dorsal muscles and elytrophore muscles in the dorsal part of the parapodium gives space for extensive branchia (electronic supplementary material, figure S5A), which are absent in the other species. The elytrophore muscles (#XX) are more numerous and distinct and the supportive dorsal muscle around it is prominent (figure 5). Noticeable is the great proportion of interparapodial bracing muscles (#VII, figure 4).

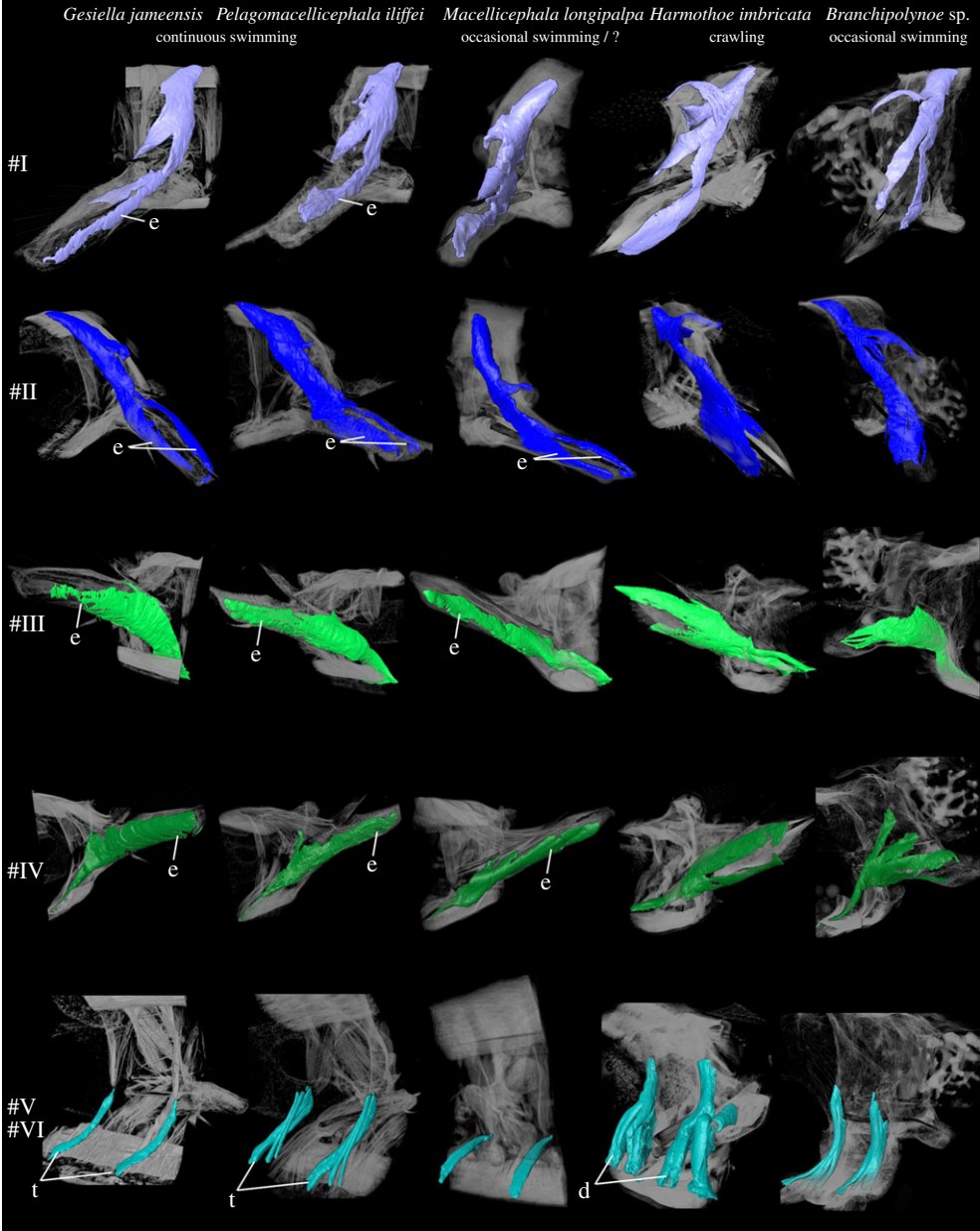

**Figure 3.** Comparison of muscle groups #I–VI for *Gesiella jameensis*, *Pelagomacellicephala iliffei*, *Macellicephala longipalpa*, *Harmothoe imbricata* and *Branchipolynoe* sp. Volume rendering based on microCT data from mid-parapodium with aciculae (pipe structure, black). Dorsal is up in all images. Colour coding and numbering; #I, anterior dorsal muscle, blue; #II, posterior dorsal muscle, dark blue; #III, anterior ventral muscle, green; #IV, posterior ventral muscle, dark green; #V and VI, oblique muscles, blue-green. Letters indicate characteristic muscles with special adaptation to locomotion. Abbreviations; d, dense; e, elongation; t, thin.

## 3.2. Parapodial muscle volumes

Muscle volumes were extracted from each parapodial reconstruction of the 21 larger muscle groups (table 3; electronic supplementary material, figure S1). *Harmothoe imbricata* had the greatest total parapodial muscle volume relative to segment volume (corr., 0.28), and *G. jameensis* had the lowest total parapodial muscle volume relative to segment volume (corr., 0.11) (table 3). The relative parapodia length was significantly different between species (ANOVA tests: $p < 0.001$) and similar within species (*t*-tests: $p < 0.05$), except for *M. longipalpa* (insufficient specimens). Dorsal and ventral muscles (#I–IV) dominated the musculature by volume, with similar relative volumes of 12.4 up to 17.6%, except in *H. imbricata* that had both lower relative volumes (6.1%–9.1%) and no appreciable difference from those of oblique, acicular and elytrophore muscle groups (#V and VI, XII, XVI, XX).

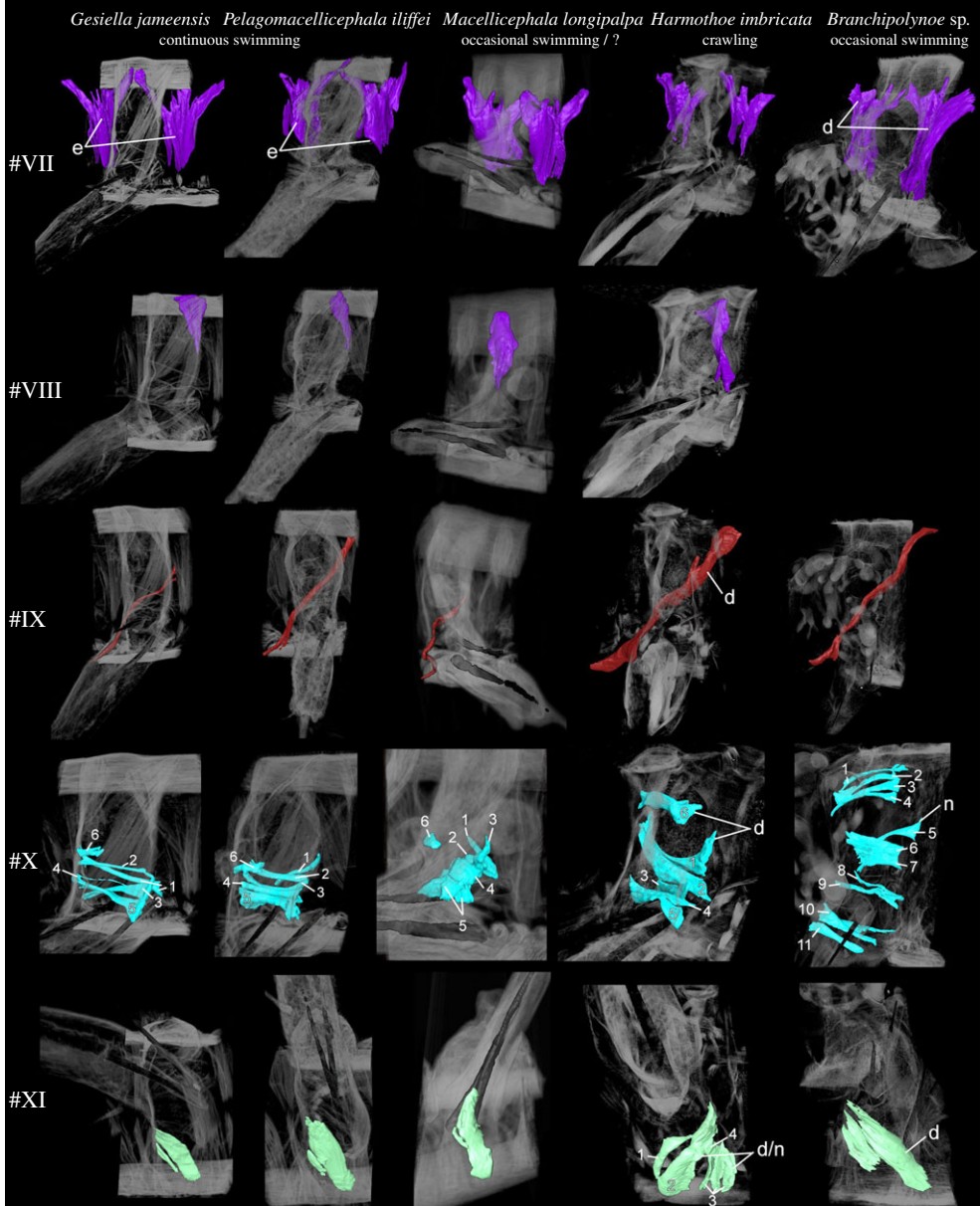

**Figure 4.** Comparison of muscle groups #VII–XI for *Gesiella jameensis*, *Pelagomacellicephala iliffei*, *Macellicephala longipalpa*, *Harmothoe imbricata* and *Branchipolynoe* sp. Volume rendering based on microCT data from mid-parapodium with aciculae (pipe structure, black). Dorsal is up in all images except for muscle #XI, which is shown in ventral view. Muscle group #VIII is not included in *Branchipolynoe* sp. since it is merged with muscle #VII. Numbers 1–11 are individual muscle bundles. Colour coding and numbering; #VII, preceding and succeeding bracing muscles, purple; #VII, additional bracing muscles, purple; #IX, dorso-ventral crossing muscle, dark red; #X, anterior-posterior transverse muscles, turquoise; #XI, ventral diagonal muscles, lime green. Letters indicate characteristic muscles with special adaptation to locomotion. Abbreviations; d, dense; e, elongation; n, numerous.

Bracing muscles (#VII and VIII) constituted high relative volumes in all species (up to 25.6% in *Branchipolynoe* sp.), except for the benthic crawling *H. imbricata* which had limited bracing muscles, constituting only 5.9% of the total volume (electronic supplementary material, table S1). *Pelagomacellicephala iliffei*, *G. jameensis*, *Branchipolynoe* sp. and *M. longipalpa* showed similar relative volumes to each other, especially in dorsal, ventral, anterior-posterior transverse and acicular/chaetal muscles, but differed in relative volumes of neuropodial chaetal sac and elytrophore muscles. *Harmothoe imbricata* was distinct from the others by having a more equal distribution of relative volumes across the different muscle groups. This shallow water species had the largest proportion of oblique, anterior-posterior transverse and elytrophore muscles with the most noteworthy difference being the oblique, acicular and chaetal muscles. No interacicular muscles were found in *P. iliffei*, *G. jameensis* or *M. longipalpa*.

**Table 3.** Parapodium morphometric overview. The table shows actual total muscle volume in mm³ (left) and the size correlated volume (muscle volume value relative to segment volume, corr.) (right), number of individual muscle bundles (left) and muscle groups (right) and relative parapodia length (parapodia length: body width ratio) for *Gesiella jameensis*, *Pelagomacellicephala iliffei*, *Macellicephala longipalpa*, *Harmothoe imbricata* and *Branchipolynoe* sp. ANOVA tests for relative parapodia length between each species showed a *p*-value < 0.001. [a]*p*-value < 0.05 (*t*-tests for relative parapodia length between specimens of each species).

| species | total muscle volume (mm³)/(corr.) | total individual muscle bundles/muscle groups | relative parapodia length |
|---|---|---|---|
| *Gesiella jameensis* | 0.0439/0.1108 | 51/20 | 1.74[a] |
| *Pelagomacellicephala iliffei* | 0.0347/0.1340 | 52/20 | 1.4[a] |
| *Macellicephala longipalpa* | 1.2026/0.2474 | 51/20 | 1.56 |
| *Harmothoe imbricata* | 0.1511/0.2826 | 67/21 | 0.81[a] |
| *Branchipolynoe* sp. | 0.7215/0.1280 | 61/20 | 1.12[a] |

[a]*p*-value < 0.05.

## 3.3. Locomotory movements of parapodia during swimming

The locomotion of *P. iliffei*, *G. jameensis* and *H. imbricata* are described from video of live specimens; however, recordings of *M. longipalpa* and *Branchipolynoe* sp. were not performed because we did not have access to live specimens.

Cave species *G. jameensis* and *P. iliffei* are only capable of swimming forward but exhibit forward and backward crawling when confined in containers. However, crawling should not be considered a natural behaviour since it has only been observed in the laboratory. It is not their preferred locomotion, and in the laboratory, they would only crawl short distances as an intermediate state before swimming. Their crawling pattern is relatively similar to that of the benthic *H. imbricata,* which is described below. Both cave species exhibit a similar swimming style, with *in situ* and *in vitro* videos showing that both species are highly manoeuvrable with a great diversity of movements. Video recordings of swimming cave species (in dorsal and lateral views) demonstrate that they swim using a combination of body undulations and parapodial power and recovery strokes. Body undulation wave velocity is increased with a higher swimming speed. The power and recovery strokes can be summarized in four steps: 1: The power stroke is initiated when the parapodium is at an anterior angle of approximately 150° to the lateral body wall. From this angle, the parapodium is moved posteriorly until it contacts the posteriorly adjacent parapodium. 2: The parapodium gathers with the previous power stroke-performing parapodia. 3: The recovery stroke is initiated when 5–6 parapodia of one power stroke-series are gathered in a cluster. The parapodia are then moved forward (from posterior to anterior but with a phase delay along the body) in a forward-directed metachronal wave. 4: During the wave, the posteriormost parapodia stops when it reaches approximately 150° in the anterior direction, where it is positioned again for the power stroke and the cycle repeats (figure 7). The wave on one side is complemented by an antagonal wave on the opposite side. During these swimming steps, neither elevation nor depression of parapodia was observed, but it cannot be ruled out.

The detailed camera set-up used exclusively in *P. iliffei* recordings showed that the chaetae spread wider (approximately up to 50% more surface area) during the power stroke than the recovery stroke but more detailed quantitative data were unfortunately not obtainable. *In situ* recordings of *G. jameensis* and *P. iliffei* showed that both spread their cirri and orient their parapodia ventrally when they are drifting. When disturbed, their dorsal cirri are folded against the body as swimming is initiated (figure 8).

Specimens of *H. imbricata* are able to crawl forward and backwards and to swim forward for short durations. Camera recordings in lateral view showed that crawling can be summarized in five steps. Considering a single parapodium during forward crawling: 1: Power stroke begins when the parapodium is facing approximately 150° anterior angle from the body line. It is then depressed towards the ground and moved posteriorly. When the parapodium is perpendicular to the body line, the body is raised above the substrate. 2: The parapodium continues posteriorly until it reaches approximately 30° in the posterior direction, pushing the body forward. 3: The posterior facing parapodium is lifted from the ground. 4: The lifted parapodium gathers with the succeeding

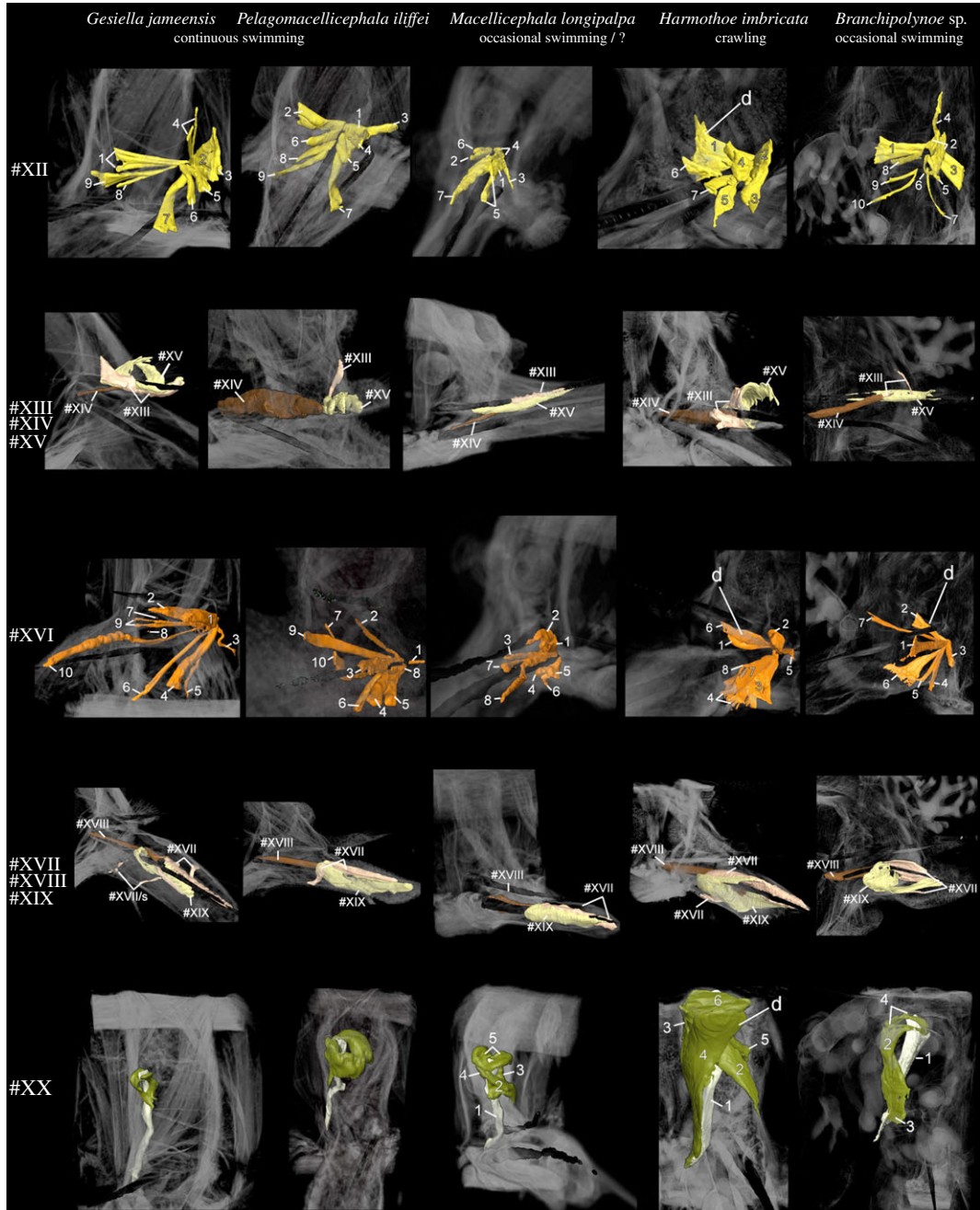

**Figure 5.** Comparison of muscle groups #XII–XX for *Gesiella jameensis*, *Pelagomacellicephala iliffei*, *Macellicephala longipalpa*, *Harmothoe imbricata* and *Branchipolynoe* sp. Volume rendering based on microCT data from mid-parapodium with aciculae (pipe structure, black). Dorsal is up in all images. Numbers 1–10 are individual muscle bundles. Colour coding and numbering; #XII, notopodial acicula muscles, yellow; #XIII, notopodial protractor muscles, pink; #XIV, notopodial retractor muscle, brown; #XV, notopodial chaetal sac muscle, beige; #XVI, neuropodial acicula muscles, orange; #XVII, neuropodial protractor muscles, pink; #XVIII, notopodial retractor muscle, brown; #XIX, neuropodial chaetal sac muscle, beige; #XX, elytrophore muscles, olive green, (depressor muscle, egg white coloured). Letters indicate characteristic muscles with special adaptation to locomotion. Abbreviations; d, dense; s, suspensor muscles.

parapodia, which initiates the recovery stroke. Together the parapodia are promoted anteriorly in a wave with phase delay until it reaches 150°. 5: The anterior facing parapodium breaks with the wave and relaxes, while the parapodium is lowered to the ground and followed by step 1 again (figure 7). No obvious body undulations are observed during crawling. *Harmothoe imbricata* is capable of forward swimming, but only does so over short distances and the body is only lifted a few millimetres above the substrate. While swimming, *H. imbricata* gains higher velocities than while crawling and body

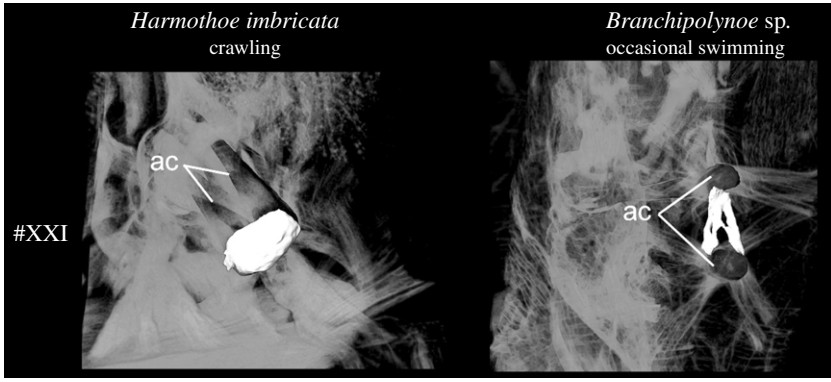

**Figure 6.** Comparison of the interacicular muscle #XXI for *Harmothoe imbricata* and *Branchipolynoe* sp. Volume rendering based on microCT data from mid-parapodium with aciculae (pipe structure, black). Dorsal is up in all images and shown from proximal view. Muscle #XXI is lacking in *Gesiella jameensis* and *Pelagomacellicephala iliffei*. Colour coding and numbering; #XXI, interacicular muscle, white. Abbreviations; ac, acicula.

**Figure 7.** Locomotion of *Gesiella jameensis*, *Pelagomacellicephala iliffei* and *Harmothoe imbricata* in laboratory set-up; (*a*) shows video frames of swimming *P. iliffei* (i), *G. jameensis* (ii) and *H. imbricata* (iii) in dorsal view. (*a*,*b*). Numbers 1–4 refer to parapodia movement during swimming (see results, §4.3). (*b*) Showing the video frame of swimming *P. iliffei* in lateral view. (*c*) Frame shot of crawling *H. imbricata* in lateral view. Numbers 1–5 refer to parapodial movements during crawling steps (see results, §4.3). Scale bars are made by estimates from fixed material.

undulations are initiated; yet the velocity gained during swimming is less than that of *G. jameensis* and *P. iliffei*. *Harmothoe imbricata* overall exhibits similar swimming patterns to *G. jameensis* and *P. iliffei* but parapodia are much stiffer and the parapodia are much closer together when performing power and

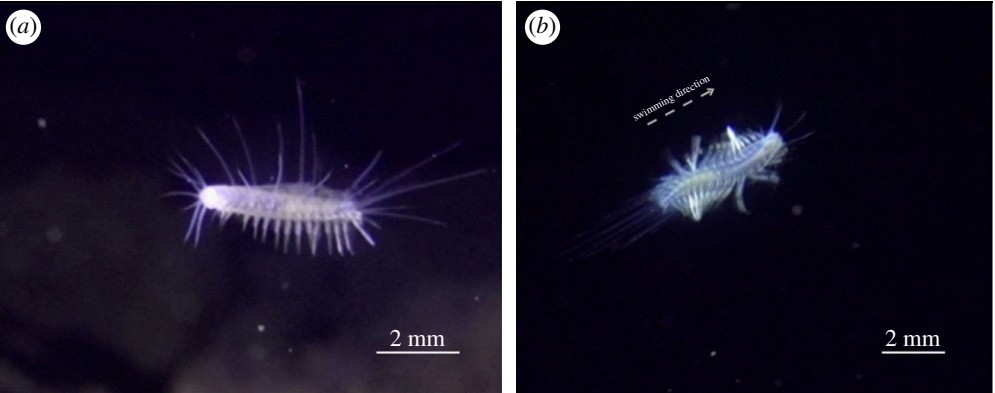

**Figure 8.** *In situ* frame shots from *Gesiella jameensis* within Cueva de los Lagos, Lanzarote, Canary Islands, Spain. (*a*) *Gesiella jameensis* motionless with spread dorsal cirri and downward hanging parapodia. (*b*) Swimming behaviour of *G. jameensis* after being disturbed by divers. Scale bars are made by estimates from fixed material.

recovery strokes. *Harmothoe imbricata* does not seem to possess the same horizontal and vertical flexibility of the body and manoeuvrability as *P. iliffei* and *G. jameensis* (figure 7).

## 4. Discussion

This study provides insight into the myoanatomical adaptations of locomotion in scale worms by comparing parapodial musculature and behaviour among crawling, occasional swimming and continuous swimming members of the subfamilies Polynoinae, Macellicephalinae and Lepidonotopodinae.

### 4.1. Muscle groups comparison between crawling and continuous swimming species

The longitudinal body wall muscles are important for propulsion in all species and do not show clear differences between species. However, the parapodial musculature varies and key differences between crawlers and swimmers can be summarized as follows: continuous swimmers have longer and thinner dorsal and ventral parapodial muscles (#I–IV, blue and green, figures 1 and 2), lack the interacicular muscle (#XXI, white, figures 1 and 2) and have thinner and more elongated bracing muscles (#VII and VIII, purple, figures 1 and 2) than the crawling species. Crawling species, however, exhibit much more voluminous oblique muscles (#V–VI, blue-green, figures 1 and 2) and thicker and shorter acicular and chaetal muscles (#XII–XIX, orange, brown, salmon colour, beige and yellow, figures 1 and 2), and thicker and more numerous elytrophore muscles (#XX, olive green, figures 1 and 2).

The parapodial myoanatomy of the swimming *G. jameensis*, *P. iliffei* and *M. longipalpa* is quite similar in terms of number of bundles, structure, muscle proportions and point of origin/insertion. This may not only reflect common locomotory patterns, but the fact that they all belong to the subfamily Macellicephalinae, within which the cave genera *Gesiella* and *Pelagomacellicephala* are found to be sister groups [23]. Despite the elongation and the reduced number of muscle bundles, the myoanatomy is also directly comparable to that found in *Branchipolynoe* sp. and *H. imbricata* (figure 2; electronic supplementary material, figures S2–S5). The oblique and bracing body wall musculature and the dorsal, ventral, acicular and chaetal muscles of the parapodia are directly comparable to prior studies of scale worms and other errantian annelids [8–10,42,45]. Even in Onychophorans, dorsal and ventral muscles (#I–IV) with points of origin and insertion much like that found in annelids are reported. Onychophorans even have intrinsic protractor and retractor muscles of the claw that are comparable to the chaetal muscles in annelids [46]. Despite these similarities, there are some differences in the detailed morphology of some muscle groups with regard to form, number and volume among the examined scale worms.

### 4.1.1. Elongation of dorsal and ventral muscles (#I–IV) in swimming species

The dorsal anterior and posterior muscles (#I and II) and ventral anterior and posterior muscles (#III and IV) originate, respectively, from the dorsal and ventral longitudinal muscle bands and extend throughout the parapodia and terminate at the distal end of the parapodium (figure 1). The muscles likely serve a key role in the oaring movements that create the power and recovery strokes [8,46,47] that in contrast with

other errant annelids (e.g. *Nereis*), these muscles constitute most of the muscular volume in scale worms [8,9,11,42,45].

Although the dorsal and ventral muscles (#I–IV) still dominate in relative volume, we found that these muscle groups are more elongated and thinner in the swimming species *G. jameensis*, *P. iliffei* and *M. longipalpa* (and presumably in *Drieschia* sp.) than in the predominantly crawling *H. imbricata* and *Branchipolynoe* sp. Myoanatomical studies of the occasional swimmers *Nereis* and *Nephtys* (Nephtyidae) likewise show elongated and thinner dorsal and ventral muscles, which are known to perform a similar function during locomotion [8,48].

### 4.1.2. Thicker and shorter acicular and chaetal muscles (#XII–XIX) in crawling species

The acicular and chaetal muscles (#XII–XIX) supporting the chaetae are a part of the intrinsic parapodial musculature. The acicular muscles originate at the proximal end of the aciculae and extend closely around the acicula into both the notopodium and neuropodium, whereas the chaetal muscles originate at the proximal part of the notopodium and neuropodium and terminate at the distal part of their body wall (figure 1). They are likely associated with locomotion and protection (deter predation by splaying their chaetae) and are well-developed in the examined benthic crawling species [5,10,42]. In *H. imbricata*, these muscles take up 26.9% (compared to 9.4–15.9% in the other examined species) of the total parapodial muscle volume (figure 2; electronic supplementary material, table S1). Likely, the stout and numerous chaetae in *H. imbricata* need further muscular support for extension, retraction, elevating and depression when crawling through mud and sand [11,12,42]. By contrast, swimming species would only need to control the spread of the chaetae to optimize for maximum drag during power and minimum drag during recovery strokes and possibly to stabilize or stiffen them during power strokes.

### 4.1.3. Interacicular muscle (#XXI) lost in continuous swimming species

The interacicular muscle (#XXI) connects the proximal ends of the noto- and neuroacicula and is positioned at the proximal end of the parapodium (figure 1). While few papers address the function of the interacicular muscle, both [10] and [42] indicate that they help the parapodium resist the drag and pull involved in crawling or digging through sediment by functioning as an antagonist in the movement of the aciculae. In pelagic and continuous swimming annelids, the parapodium rarely comes in contact with the substrate and therefore does not need the muscular support of the aciculae. Interestingly, the interacicular muscle (#XXI) is lacking in the highly mobile and swimming species examined (*G. jameensis* and *P. iliffei*) as well as in the incomplete reconstructions of *Drieschia* sp. Interestingly, interacicular muscles are also lacking in the genus *Nephtys*, but earlier parapodial investigations observed them in other benthic, crawling species of Aphroditidae, Polynoidae, Sigalionidae and Nereididae [5,9]. The holopelagic *Tomopteris* lacks aciculae completely, likely to enhance parapodial flexibility even further. Extra muscles associated with the aciculae may even decrease flexibility and movement of the parapodium [11]. In Sedentaria families like Terebellidae and Scalibregmatidae, these muscles are further connected with muscle fibres that originate at the ventral midline [5,9,11], presumably aiding the anchoring of their chaetae in tubes and burrows.

### 4.1.4. Reduction of bracing muscles (#VII and VIII) in swimming species

The bracing muscles support the body wall and are found intersegmentally, between the parapodia, branching anterior and posteriorly towards the adjacent parapodia (figures 1 and 4). It has been suggested that the lack of circular muscles in many errant annelids is compensated for by bracing muscles [5,11,49]. The bracing muscles are normally associated with maintaining the body wall constrictions and peristaltic movement [5]. In *Branchipolynoe* sp., these muscles have the greatest observed proportions and take up 25% of the total muscle volume (electronic supplementary material, table S1). For *Aphrodita aculeata*, the bracing muscles are more complex and comprise three rows of dorsal, lateral and ventral muscles [42], likely supporting the additional parapodial strength required when burrowing and moving in the sediment. In other annelids, the bracing muscles are even more diverse and varied, with large lateral crossed bundles in *Nereis*, to thin dorsal fibres in *Dorvillea* (Dorvilleidae) [5,8,11]. The greater proportion of bracing muscles likely supports crawling and aids in twisting, bending and constriction of the body. Given the broad (or well-developed) bracing muscles in *Branchipolynoe*, we interpret this as an adaptation for manoeuvring in and out of mussels and around rugged mussel beds.

### 4.1.5. Reduced elytrophore muscles (#XX) in swimming species

These muscles are only found in scale worms since they are the only annelids to possess elytra. However, this musculature may be homologous to that of the cirrophore of the dorsal cirrus, a structure found in many other annelids. Both cirrophores and elytrophores are dorsally positioned close to the dorsal muscles (#I and II) (figure 1). Cirrophores and elytrophores comprise extrinsic and intrinsic muscles to depress and levitate either the elytra or the sensing style of the dorsal cirrus [42]. The elytrophore muscle bundles are both more numerous and denser in the benthic *H. imbricata*, having six bundles (figure 5), than in the continuous swimming cave species *G. jameensis* and *P. iliffei* only possessing an elytrophore depressor muscle and a supportive elytrophore wall musculature (figure 2; electronic supplementary material, figure S2). These muscular reductions in the cave species likely reflect the otherwise protective and ventilating roles of the elytra in open water, benthic species. *Harmothoe imbricata*, living in shallow water habitats, is capable of illuminating its elytra upon disturbance and even shedding them when attacked to further distract its decapod and fish predators [10,50]. In caves where larger predators are either entirely lacking or visually limited, scale worms like *G. jameensis* and *P. iliffei* are among the top predators [4,51] and fewer muscles are allocated to the elytrophore. The elytra of these cave scale worms are also thinner and lighter than other scale worm elytra, which would require fewer muscles and their reduction possibly further adds to an increased buoyancy of these swimming species [34].

## 4.2. Behavioural and muscular adaptations to a continuous swimming lifestyle in Annelids

Our video recordings show that all examined species were able to swim by the use of parapodial power and recovery strokes in combination with body undulations (figure 7). However, when examining their swimming behaviour in detail, the degree of swimming ability varies greatly [17,19,22,52]. Even though *H. imbricata* can swim, it only hovers a few millimetres off the substrate for short distances, and primarily moves by crawling. Videos of *Lepidonotus*, another shallow water scale worm, show similar swimming just above the sediment [53]. In the opposite extreme, both *G. jameensis* and *P. iliffei* move exclusively by swimming and drifting in the water column. Their drifting is likely aided by the observed outstretching of their dorsal cirri, reducing their sinking rate by potentially increasing drag (figure 8*a*). However, during the collection of *G. jameensis*, we observed that their escape response was to head into cave cracks and crevices, which required crawling. *Gesiella jameensis* and *P. iliffei* both have excellent swimming abilities and show similar parapodial movements to the holopelagic *Tomopteris* [15,18,22,54]. Furthermore, the chaetae of *P. iliffei* spread wide during power stroke and fold again during recovery stroke, most likely to increase the chaetal surface area and thus drag during power stroke. This is likely important for effective swimming at relatively intermediate Reynolds similar to pinnule spreading in *Tomopteris* [15].

The elongated parapodia observed in *G. jameensis* and *P. iliffei* most likely reflect their relationship within Macellicephalinae rather than a new adaptation. However, this may be an apomorphy and adaptation of this entire deep-sea subfamily since it is a typical trait of many of its members, most of which are also assumed to be epibenthic and capable of swimming but have yet to be observed doing so. Similarly, elongated parapodia are widespread across holopelagic annelids, including all members of Tomopteridae, Alciopini and Lopadorrhynchidae. These parapodia evidently function as 'oars' and their surface area provides greater thrust and possibly helps achieve greater velocity and efficiency [15,55]. The greater dimension of such parapodial elongation would increase the muscle output and potentially the velocity. Although not quantitatively assessed, our video observations support this hypothesis, showing relatively faster parapodial movements in the continuous swimming species. The proposed myoanatomical features associated with continuous swimming, elongated parapodia muscles and fewer and thinner muscle bundles were also observed in *M. longipalpa* (Macellicephalinae). Based on our observations of the high parapodial velocity in *G. jameensis* and *P. iliffei*, this suggests that *M. longipalpa* is a capable swimmer, however, its total and relative parapodial muscle volume is similar to that of *H. imbricata* (table 3) and likely prevents it from staying in the water column when not actively swimming [39,56].

*Branchipolynoe* sp. exhibits a musculature that shares total relative parapodial muscle volume, elongated extrinsic dorsal and ventral muscles and thin acicular chaetal muscles with the swimming species *G. jameensis* and *P. iliffei*, and numerous stout musculature with the crawling *H. imbricata*. The swimming style of *Branchipolynoe* sp. may therefore be more explosive, as seen in *H. imbricata*, due to

greater force production. However, it will likely be able to stay longer in the water, which fits well with its reported benthopelagic lifestyle and periodic swimming between bivalve hosts [17].

Crawling does not require long parapodia since it relies on contact with the substrate and movements of strong depressor, retractor and protractor muscles for lift and forward motion. In a benthic environment, it is thus likely that short parapodia with numerous, dense muscle bundles and high relative muscle volume, as documented here for *H. imbricata* and *Branchipolynoe* sp., are beneficial when crawling [10,46,47,57]. A similar muscle arrangement is also found in other benthic crawling and digging scale worms i.e. *Lepidonotus squamatus* and *A. aculeata*, the latter possessing even more complex and compact musculature [9,42]. Even though data on parapodial musculature in sedentarian annelids are scarce, the few studies available suggest that muscularized septa and extra acicular muscle connections to the ventral midline are important for enhanced efficiency in burrowing [5,9].

Reduced muscle number, density and relative volume are documented here for the cave worms *G. jameensis* and *P. iliffei* and indicated for the holopelagic *Drieschia*. Buoyancy will generally increase with reduced muscle volume if replaced by body fluid similar in density to seawater. This is best observed in Tomopteridae where their myoanatomy has been reduced to a thin mesh just below the epidermis, leaving a large fluid-filled body cavity [58]. In addition to reduced weight, the loss of some muscle groups (e.g. the interacicular muscles as documented here) [10,42] may increase parapodial flexibility. Increased parapodial flexibility may be further supported by the development of other muscle groups such as elongated parapodial musculature [5,9,12,59]. Notably, these features are repeated in each segment and thus provide an overall reduction of weight and density of the entire animal.

# 5. Conclusion

This study has provided insight into the myoanatomy associated with locomotion in scale worms. By comparing different scale worms and lifestyles, common muscle groups were identified and their function predicted. Differences in parapodial musculature were also detected, which allowed for the identification of adaptations to crawling and swimming, respectively.

Adaptations for continuous swimming are a reduction in the number, density, volume, as well as a lengthening of parapodial muscles, as seen in *G. jameensis* and *P. iliffei*. Both the elongated parapodia and dorsal cirri that provide a larger surface area to volume ratio and the fewer, thinner muscles that decrease body mass and density, increase buoyancy and their ability to passively maintain their position in the water column during swimming and drifting. These results showed that even though polynoids exhibit relatively similar muscle arrangements in terms of point of origin and insertion, the locomotion in crawling and swimming species is mediated by different muscle groups. In swimming species, the dorsal and ventral muscles #I–IV likely serve a key role in swimming and oaring movements of the parapodia, and elongation of these muscles is necessary for supporting their longer parapodia. In crawling annelids, well-developed acicular and chaetal muscles are highly important when moving through sediment [11,42]. The interacicular muscle is absent in species with continuous swimming abilities and its function must be mainly to support the position of the acicula during crawling locomotion. However, short swimming forays are still conceivable and common in benthic, crawling scale worms. Our locomotory observations reveal that benthic crawling species have similar swimming patterns to those of continuously swimming species, which likely are also supported by similar (though shorter) dorsal and ventral parapodial muscles as well as longitudinal body wall muscles.

Through morphological and behavioural comparison between different scale worms and their locomotion, we are able to suggest morphological changes associated with accessing the water column. However, continuous swimming species were only represented here by the two closely related cave-dwelling species, and many of their adaptations may reflect a common origin. Our preliminary study of *Drieschia*, although the material was very limited, suggested that highly adapted swimming annelids (e.g. *Drieschia* and tomopterids) may have undergone even further (and likely convergent) reduction of parapodial muscles. Morphological studies of these holopelagic annelids would help us understand the extremes of the scale worm swimming adaptations, provide a unique lineage for comparison and better document the adaptive nature of annelids as a whole.

**Ethics.** All applicable international, national and/or institutional guidelines for animal testing, animal care and use of animals were followed by the authors. The experiments in this study did not require an approval by an ethical committee. All procedures in this investigation comply with international and institutional guidelines, including

the guidelines for animal welfare. Animals were collected under the scientific research permit (SRP # 18-12-01-23) issued by the TCI department of environmental and coastal resources (DECR), issued through TMI for sampling in Turks and Caicos, whereas collecting in other waters (e.g. Spain, Denmark) did not require sampling permits for segmented worms.

Data accessibility. The datasets supporting this article are accessible via Dryad repository: Allentoft-Larsen *et al.* 2021 https://doi.org/10.5061/dryad.n02v6wwws [60]. Allentoft-Larsen *et al.* 2021 https://doi.org/10.5061/dryad.ngf1vhhtg [61]. Allentoft-Larsen *et al.* 2021, https://doi.org/10.5061/dryad.gqnk98smq [62]. Allentoft-Larsen *et al.* 2021 https://doi.org/10.5061/dryad.4tmpg4f98 [63]. Allentoft-Larsen *et al.* 2021 *Branchipolynoe* sp. MicroCT Scans for three-dimensional reconstruction, Dryad, Dataset, https://doi.org/10.5061/dryad.kd51c5b5d [64]. Allentoft-Larsen *et al.* 2021 https://doi.org/10.5061/dryad.r2280gbcq [65]. Allentoft-Larsen *et al.* 2021 https://doi.org/10.5061/dry ad.0rxwdbs0b [66].

The data are provided in the electronic supplementary material [67].

Authors' contributions. K.W. designed and coordinated the study. M.C.A., K.W. and B.C.G. collected field data, carried out confocal scanning microscopy along with morphological measurements, statistical analyses, three-dimensional reconstruction, video recordings, imaging and data analysis. B.C.G. and K.O. generated microCT data. M.C.A. and K.W. drafted the manuscript. M.C.A., B.C.G. and K.W. participated in data analysis. J.D. carried out high-speed footage of *P. iliffei*. M.C.A., K.W., B.C.G., K.O., K.K. and J.D. critically revised the manuscript. All authors gave final approval for publication and agree to be held accountable for the work performed therein.

Competing interests. We declare we have no competing interests.

Funding. Field and laboratory work during the expedition to Turks and Caicos in 2019 was financially supported by The Smithsonian's Global Genome Initiative (grant no. GGI-2019-Rolling-214), the Danish Natural History Society and the Peter Buck Fellowship Program. Further laboratory work was supported by the Carlsberg Foundation (grant no. CF15-0946) and the David and Lucile Packard Foundation.

Acknowledgements. We would like to thank the members of the 'Caicos Caves III' team during the 2019 expedition to Turks and Caicos: Thomas Iliffe, Jørgen Olesen, Sarit B. Truskey, Lauren Ballou, Paul Heinerth, Mark Parrish, Naqqi Manco and Jon Ward. For collection of *Gesiella*, we thank the local Environmental Service of the Cabildo de Lanzarote, Alejandro Martínez, Malte Jarlgaard Hansen and Alvaro Garcia Herrero, all aiding with permissions and logistics to access Túnel de la Atlántida and Cueva De Los Lagos for diving and collecting samples. Thanks to Paula Mendoza (student of Katrine Worsaae) for the collection of *H. Imbricata*. We are grateful to Sofie Gudnitz-Larsen for her help and support with illustrations and to Freya Goetz for her assistance with the microCT scanning. Special thanks to Alexandra Kerbl and María Herranz for their initial support and discussion. We greatly appreciate the thoughtful suggestions of the three reviewers that helped us improve the manuscript. Lastly, Sanni Jensen deserves a great thanks for her guidance and support of the study process.

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
