## [Peer Review File · Royal Society Open Science]

Review History

RSOS-210541.R0 (Original submission)

Review form: Reviewer 1

Is the manuscript scientifically sound in its present form?

Yes

Are the interpretations and conclusions justified by the results?

Yes

Is the language acceptable?

Yes

Do you have any ethical concerns with this paper?

No

Have you any concerns about statistical analyses in this paper?

No

Recommendation?

Accept with minor revision (please list in comments)

Comments to the Author(s)

This is an interesting study of the potential morphological changes associated with swimming in annelid worms. The authors studied the morphology and movements of five scale worm species that show a range of locomotion including the typical benthic crawling of annelids, occasional swimming, and continuous swimming. The morphology of the musculature of the parapodia was investigated with confocal laser scanning microscopy and microCT. The microCT data were used in 3D reconstruction and in estimates of muscle volume. Video recordings of swimming were made in several of the species for which live animals were available. The morphology of the musculature was described, along with estimates of the volume of the various muscles. Basic aspects of the swimming movements were described from the video recordings. The results were discussed in the context of attempting to identify specializations associated with swimming in annelids.

The study provides a detailed description of the numerous (21) muscles of the parapodia in the five species, taking good advantage of the remarkable morphological data provided by microCT methods. The 3D reconstructions of the muscles included in the figures are nicely presented, help in understanding the rather complicated morphology, and represent a significant contribution to our understanding of annelid muscular morphology. Although the individual panels in the figures are a bit small, since the majority of readers will view the figures online and can adjust the size on screen, I do not believe that the size will be a problem, especially since readers can consult the more extensive morphological descriptions and excellent figures in the supplemental materials.

The manuscript overall is quite well written and clear. The interpretation of the role of the various muscles in locomotion is reasonable, but it is necessarily limited by the fact that most of the techniques (electromyography, sonomicrometry, muscle mechanics, x-ray imaging, strain gauge measurement, etc.) are not feasible on these relatively small specimens. I have included a few minor comments and suggestions below.

Page	Line	Comment
2	38	Revise to read: "... but are negatively buoyant and thus cannot passively maintain.."
5	30	It would be helpful to include the internal diameter of the pipette tip here to provide some sense of the volume scanned. I suspect that many readers will be like me and not be familiar with the size of the species described in the study. In general it would be helpful to add scale bars to the 3-D reconstruction figures and the live animal images.
10	37	As is mentioned later in the paper, the reduction in drag of the parapodia on the return stroke is critical to providing a net thrust for swimming. It would thus be of interest to provide a more detailed description here on your observations of the folding and unfolding of the chaetae during the return and power strokes. Did you note differences between the species studied? This issue of the relative drag of the two strokes is particularly important given the small size, and thus relatively low Reynolds Number that these animals experience. It would also be of interest to note any differences in the relative duration of the power and return strokes. At high Reynolds numbers a slower return stroke relative to the power stroke can provide net thrust, but this works less and less well as Reynolds numbers decrease; at low Reynolds numbers reconfiguration to reduce drag on the return stroke is essential.
14	20	Revise to read: "...support crawling and aid in twisting, bending, and constriction..."

14 52,54 Replace "less" with "fewer"

15 8 It might be worthwhile to consider the implications of the elongation of the muscles for the output of the muscles in swimming. Because of the implications of series vs parallel elements in muscle, the output of a muscle depends greatly on its dimensions. For example, compare two muscles that are identical in fiber type and in volume but muscle "A" is twice as long as muscle "B". The shortening velocity of muscle A will be twice that of muscle B. But muscle B will have a greater cross-sectional area (because we are holding volume constant in this case) and thus its peak isometric tension will be twice that of muscle A. Interestingly, since power is force times velocity the two muscles will have the same power output. This is also why I made the comment above about providing additional detail on the velocity of the power and return strokes. For instance, are the power strokes of the parapodia of the swimming animals faster than those of the benthic crawlers? The elongated muscles of the swimming animals would facilitate this. The stouter muscles of the crawlers would be capable of greater force production, assuming that they do indeed have a larger relative cross-sectional area.

Review form: Reviewer 2

Is the manuscript scientifically sound in its present form?

Yes

Are the interpretations and conclusions justified by the results?

Yes

Is the language acceptable?

Yes

Do you have any ethical concerns with this paper?

No

Have you any concerns about statistical analyses in this paper?

No

Recommendation?

Accept with minor revision (please list in comments)

Comments to the Author(s)

Dear authors, your manuscript represents a well-written, detailed and highly needed comparative investigation of an understudied character complex in annelids. The data is well-presented and illustrated, results are carefully described and the conclusions are reasonable and embedded in recent knowledge. I therefore recommend publication of this nice contribution - there are just few minor points that should be addressed before final acceptance (see Appendix A).

Review form: Reviewer 3 (Kelly Dorgan)

Is the manuscript scientifically sound in its present form?

Yes

Are the interpretations and conclusions justified by the results?

Yes

Is the language acceptable?

Yes

Do you have any ethical concerns with this paper?

No

Have you any concerns about statistical analyses in this paper?

Yes

Recommendation?

Major revision is needed (please make suggestions in comments)

Comments to the Author(s)

This article provides an interesting and novel look at the adaptations required for polynoid annelids to develop a continuously swimming lifestyle. The authors convincingly show that the parapodial musculature of continuous swimmers differs from those of crawlers or burrowers in ways that make sense for these different lifestyles, and that differences occur in the relative size of muscles rather than through changes in the types of muscles or in locomotion in the individuals examined. The manuscript contains a lot of really interesting data, and my comments are primarily focused on clarifying what is, on first read, an overwhelming amount of information, to clearly convey the really interesting findings of this paper.

(note that it's a bit confusing because the author page numbers are different from the pdf page numbers and the line numbers don't match the text – I'm using the author page numbers and approximate line numbers)

1. The introduction lacks a clear hypothesis. The goals currently presented (page 4 lines 22-27) are overly general and do not explicitly state the theory being tested, making it difficult to understand the reasoning behind the myoanatomical comparisons made in the results and discussion. From reading the paper, it seems like there are two clear hypotheses that are not necessarily mutually exclusive and are both very interesting – (1) the transition from crawling to swimming involves changes in locomotory behavior, specifically of the parapodia, and the underlying musculature versus (2) that the transition to swimming involves dealing with the problem of buoyancy, in which case the (relatively dense) muscle volume should be reduced in swimmers. The findings primarily support #2 but with some interesting differences in specific muscles consistent with #1.
2. The Figure 1 schematic is really helpful, and would be stronger if it were integrated into more explicit hypotheses, e.g., which muscles would one expect to be larger or smaller in swimmers versus crawlers? There's a lot of information in this manuscript, and while the muscle groups are complex and appropriately named, maybe they could be further categorized by predicted differences? I do really like the consistent use of color across the figures to keep track of which muscles are which.
3. Comparison of species within a family needs to be put in a phylogenetic context. For example, are the two swimming species closely related, or did swimming evolve more than once within the family? A figure showing the phylogeny of the examined worms (and relevant traits) would be extremely useful.

4. While the methods section was technically sufficient, it lacked justification and it was not clear how each section addressed the various goals of the research. For example, even after reading the results and discussion, it was not apparent how the confocal laser scanning microscopy contributed to this research. This could be easily addressed after the addition of hypotheses, by linking each of the various methods to the individual hypothesis that were tested. I suggest adding a sentence at the beginning of each paragraph to give a brief overview of why the methods were used before getting into the technical details. The video analysis section focuses on the video recording methods and does not actually discuss any analyses of videos other than the last sentence which mentions the software but not what was actually measured or characterized.

5. When calculating the size correlated measure of muscle volume (page 6 lines 25-27), the authors divided muscle volume by segment width, resulting in a measure of area which is difficult to interpret and does not allow for comparison across different sized species. It would be more appropriate to present a non-dimensional measure of volume by dividing muscle volume by the volume of the entire corresponding segment (or some proxy of volume, e.g., length x width x thickness). This non-dimensional metric of muscle volume could then directly test the hypotheses.

6. The statistics are somewhat confusing. I am uncertain what measurements are being compared in the t-test mentioned on page 6 lines 27-32. The results of this test are not mentioned anywhere other than in table 3 caption, and a more thorough explanation should be added. Table 3 gives p-values, but for what comparison?

7. Figures 3-6 show the differences in muscle bundle size for each of the species examined, however the figures are a bit overwhelming because the comparisons / conclusions the reader should draw are not immediately apparent. These figures would be greatly improved if they were either constrained to only the muscle bundles that illustrate the differences between a swimming and crawling lifestyle, or edited to draw more attention to the relevant comparisons. Perhaps consider moving the full figures to the supplement and making one figure that highlights the most interesting comparisons to include in the main text? It would be helpful to indicate with the species names which are crawlers and which are swimmers so the reader doesn't have to flip back to earlier sections to recall. Alternately, focus more on two species, one crawler and one swimmer, and put much of the data from the other 3 species in the supplement to simplify the message.

8. I found that comparing Figure 2 to the supplementary figure S3 really clearly illustrated the difference in muscle volume between the swimming and crawling worms. I suggest moving one of the supplementary figures into the main body of the text to allow for comparison of musculature between swimmers and crawlers.

9. I found table 4 to be too large and cumbersome to be useful. Multiple columns for each species make comparisons very difficult. A reduced table focusing on a single characteristic (I suggest normalized dimensionless muscle volume), or better yet a figure summarizing the salient points of the table would be much more useful. Table 4 should be moved to the supplement.

10. Table 2 is also fairly large and cumbersome - consider moving it to the supplement and presenting a simpler summary table or figure here. For example, consider modifying the Fig. 1 schematic to show the key differences between crawlers and swimmers, potentially focusing on one representative crawling and one swimming species.

11. Finally, the discussion makes a lot of good points, but quite a bit of the material in the discussion could be moved to the introduction in laying out clear, specific hypotheses, allowing for more elaboration on the broader implications of the work in the discussion. For example, while the reduction in muscle bundles and density shown is really interesting, how do those effects scale across the entire worm? Do these reductions reduce overall the density of the worm by an amount relevant to swimming vs crawling?

12. The title could be edited to more clearly convey the main point and reduce jargon (remove “myoanatomy”), e.g., “Reduced parapodial muscle volume facilitates swimming in Polynoidae”

Kelly Dorgan

Decision letter (RSOS-210541.R0)

Dear Dr Worsaae

The Editors assigned to your paper RSOS-210541 "Parapodial myoanatomy reveals adaptations in swimming Polynoidae (Aphroditiformia, Annelida)" have now received comments from reviewers and would like you to revise the paper in accordance with the reviewer comments and any comments from the Editors. Please note this decision does not guarantee eventual acceptance.

Please submit your revised manuscript and required files (see below) no later than 21 days from today's (ie 28-May-2021) date. Note: the ScholarOne system will 'lock' if submission of the revision is attempted 21 or more days after the deadline. If you do not think you will be able to meet this deadline please contact the editorial office immediately.

Kind regards,
Royal Society Open Science Editorial Office
Royal Society Open Science

on behalf of Dr Jonas Rubenson (Associate Editor) and Kevin Padian (Subject Editor)
openscience@royalsociety.org

Associate Editor Comments to Author (Dr Jonas Rubenson):

Comments to the Author:

Dear Dr. Worsaae,

As you will see, the reviewers are overall positive about your paper and all three reviewers find your experiment interesting. However, the reviewers raise some concerns that you will need to address upon revision. I am sympathetic to the reviewer's concern about the clarity of hypotheses. The suggestions to include a description of both power stroke and return stroke, and a description of the parapodial muscle complex are also useful.

Hopefully the reviewers' comments, which I believe are constructive, are addressable upon revision.

Reviewer comments to Author:

Reviewer: 1

Comments to the Author(s)

This is an interesting study of the potential morphological changes associated with swimming in annelid worms. The authors studied the morphology and movements of five scale worm species that show a range of locomotion including the typical benthic crawling of annelids, occasional swimming, and continuous swimming. The morphology of the musculature of the parapodia was investigated with confocal laser scanning microscopy and microCT. The microCT data were used in 3D reconstruction and in estimates of muscle volume. Video recordings of swimming were made in several of the species for which live animals were available. The morphology of the musculature was described, along with estimates of the volume of the various muscles. Basic aspects of the swimming movements were described from the video recordings. The results were discussed in the context of attempting to identify specializations associated with swimming in annelids.

The study provides a detailed description of the numerous (21) muscles of the parapodia in the five species, taking good advantage of the remarkable morphological data provided by microCT methods. The 3D reconstructions of the muscles included in the figures are nicely presented, help in understanding the rather complicated morphology, and represent a significant contribution to our understanding of annelid muscular morphology. Although the individual panels in the figures are a bit small, since the majority of readers will view the figures online and can adjust the size on screen, I do not believe that the size will be a problem, especially since readers can consult the more extensive morphological descriptions and excellent figures in the supplemental materials.

The manuscript overall is quite well written and clear. The interpretation of the role of the various muscles in locomotion is reasonable, but it is necessarily limited by the fact that most of the techniques (electromyography, sonomicrometry, muscle mechanics, x-ray imaging, strain gauge measurement, etc.) are not feasible on these relatively small specimens. I have included a few minor comments and suggestions below.

Page Line Comment

2 38 Revise to read: "... but are negatively buoyant and thus cannot passively maintain.."

5 30 It would be helpful to include the internal diameter of the pipette tip here to provide some sense of the volume scanned. I suspect that many readers will be like me and not be familiar with the size of the species described in the study. In general it would be helpful to add scale bars to the 3-D reconstruction figures and the live animal images.

10 37 As is mentioned later in the paper, the reduction in drag of the parapodia on the return stroke is critical to providing a net thrust for swimming. It would thus be of interest to provide a more detailed description here on your observations of the folding and unfolding of the chaetae during the return and power strokes. Did you note differences between the species studied? This issue of the relative drag of the two strokes is particularly important given the small size, and thus relatively low Reynolds Number that these animals experience. It would also be of interest to note any differences in the relative duration of the power and return strokes. At high Reynolds numbers a slower return stroke relative to the power stroke can provide net thrust, but this works less and less well as Reynolds numbers decrease; at low Reynolds numbers reconfiguration to reduce drag on the return stroke is essential.

14 20 Revise to read: "...support crawling and aid in twisting, bending, and constriction..."

14 52,54 Replace "less" with "fewer"

15 8 It might be worthwhile to consider the implications of the elongation of the muscles for the output of the muscles in swimming. Because of the implications of series vs parallel elements in muscle, the output of a muscle depends greatly on its dimensions. For example, compare two muscles that are identical in fiber type and in volume but muscle "A" is twice as long as muscle "B". The shortening velocity of muscle A will be twice that of muscle B. But muscle B will have a greater cross-sectional area (because we are holding volume constant in this case) and thus its peak isometric tension will be twice that of muscle A. Interestingly, since power is force times velocity the two muscles will have the same power output. This is also why I made the comment above about providing additional detail on the velocity of the power and return strokes. For instance, are the power strokes of the parapodia of the swimming animals faster than those of the benthic crawlers? The elongated muscles of the swimming animals would facilitate this. The stouter muscles of the crawlers would be capable of greater force production, assuming that they do indeed have a larger relative cross-sectional area.

Reviewer: 2

Comments to the Author(s)

Dear authors, your manuscript represents a well-written, detailed and highly needed comparative investigation of an understudied character complex in annelids. The data is well-presented and illustrated, results are carefully described and the conclusions are reasonable and embedded in recent knowledge. I therefore recommend publication of this nice contribution - there are just few minor points that should be addressed before final acceptance (see attached).

Reviewer: 3

Comments to the Author(s)

This article provides an interesting and novel look at the adaptations required for polynoid annelids to develop a continuously swimming lifestyle. The authors convincingly show that the parapodial musculature of continuous swimmers differs from those of crawlers or burrowers in ways that make sense for these different lifestyles, and that differences occur in the relative size of muscles rather than through changes in the types of muscles or in locomotion in the individuals examined. The manuscript contains a lot of really interesting data, and my comments are primarily focused on clarifying what is, on first read, an

overwhelming amount of information, to clearly convey the really interesting findings of this paper.

(note that it's a bit confusing because the author page numbers are different from the pdf page numbers and the line numbers don't match the text - I'm using the author page numbers and approximate line numbers)

1. The introduction lacks a clear hypothesis. The goals currently presented (page 4 lines 22-27) are overly general and do not explicitly state the theory being tested, making it difficult to understand the reasoning behind the myoanatomical comparisons made in the results and discussion. From reading the paper, it seems like there are two clear hypotheses that are not necessarily mutually exclusive and are both very interesting - (1) the transition from crawling to swimming involves changes in locomotory behavior, specifically of the parapodia, and the underlying musculature versus (2) that the transition to swimming involves dealing with the problem of buoyancy, in which case the (relatively dense) muscle volume should be reduced in swimmers. The findings primarily support #2 but with some interesting differences in specific muscles consistent with #1.
2. The Figure 1 schematic is really helpful, and would be stronger if it were integrated into more explicit hypotheses, e.g., which muscles would one expect to be larger or smaller in swimmers versus crawlers? There's a lot of information in this manuscript, and while the muscle groups are complex and appropriately named, maybe they could be further categorized by predicted differences? I do really like the consistent use of color across the figures to keep track of which muscles are which.
3. Comparison of species within a family needs to be put in a phylogenetic context. For example, are the two swimming species closely related, or did swimming evolve more than once within the family? A figure showing the phylogeny of the examined worms (and relevant traits) would be extremely useful.
4. While the methods section was technically sufficient, it lacked justification and it was not clear how each section addressed the various goals of the research. For example, even after reading the results and discussion, it was not apparent how the confocal laser scanning microscopy contributed to this research. This could be easily addressed after the addition of hypotheses, by linking each of the various methods to the individual hypothesis that were tested. I suggest adding a sentence at the beginning of each paragraph to give a brief overview of why the methods were used before getting into the technical details. The video analysis section focuses on the video recording methods and does not actually discuss any analyses of videos other than the last sentence which mentions the software but not what was actually measured or characterized.
5. When calculating the size correlated measure of muscle volume (page 6 lines 25-27), the authors divided muscle volume by segment width, resulting in a measure of area which is difficult to interpret and does not allow for comparison across different sized species. It would be more appropriate to present a non-dimensional measure of volume by dividing muscle volume by the volume of the entire corresponding segment (or some proxy of volume, e.g., length x width x thickness). This non-dimensional metric of muscle volume could then directly test the hypotheses.
6. The statistics are somewhat confusing. I am uncertain what measurements are being compared in the t-test mentioned on page 6 lines 27-32. The results of this test are not mentioned anywhere other than in table 3 caption, and a more thorough explanation should be added. Table 3 gives p-values, but for what comparison?
7. Figures 3-6 show the differences in muscle bundle size for each of the species examined, however the figures are a bit overwhelming because the comparisons / conclusions the reader should draw are not immediately apparent. These figures would be greatly improved if they were either constrained to only the muscle bundles that illustrate the differences between a

swimming and crawling lifestyle, or edited to draw more attention to the relevant comparisons. Perhaps consider moving the full figures to the supplement and making one figure that highlights the most interesting comparisons to include in the main text? It would be helpful to indicate with the species names which are crawlers and which are swimmers so the reader doesn't have to flip back to earlier sections to recall. Alternately, focus more on two species, one crawler and one swimmer, and put much of the data from the other 3 species in the supplement to simplify the message.

8. I found that comparing Figure 2 to the supplementary figure S3 really clearly illustrated the difference in muscle volume between the swimming and crawling worms. I suggest moving one of the supplementary figures into the main body of the text to allow for comparison of musculature between swimmers and crawlers.

9. I found table 4 to be too large and cumbersome to be useful. Multiple columns for each species make comparisons very difficult. A reduced table focusing on a single characteristic (I suggest normalized dimensionless muscle volume), or better yet a figure summarizing the salient points of the table would be much more useful. Table 4 should be moved to the supplement.

10. Table 2 is also fairly large and cumbersome – consider moving it to the supplement and presenting a simpler summary table or figure here. For example, consider modifying the Fig. 1 schematic to show the key differences between crawlers and swimmers, potentially focusing on one representative crawling and one swimming species.

11. Finally, the discussion makes a lot of good points, but quite a bit of the material in the discussion could be moved to the introduction in laying out clear, specific hypotheses, allowing for more elaboration on the broader implications of the work in the discussion. For example, while the reduction in muscle bundles and density shown is really interesting, how do those effects scale across the entire worm? Do these reductions reduce overall the density of the worm by an amount relevant to swimming vs crawling?

12. The title could be edited to more clearly convey the main point and reduce jargon (remove “myoanatomy”), e.g., “Reduced parapodial muscle volume facilitates swimming in Polynoidae”

Kelly Dorgan

===PREPARING YOUR MANUSCRIPT===

While not essential, it will speed up the preparation of your manuscript proof if accepted if you format your references/bibliography in Vancouver style (please see

<https://royalsociety.org/journals/authors/author-guidelines/#formatting>). You should include DOIs for as many of the references as possible.

===PREPARING YOUR REVISION IN SCHOLARONE===

-- Ensure that your data access statement meets the requirements at <https://royalsociety.org/journals/authors/author-guidelines/#data>. You should ensure that you cite the dataset in your reference list. If you have deposited data etc in the Dryad repository, please include both the 'For publication' link and 'For review' link at this stage.

Author's Response to Decision Letter for (RSOS-210541.R0)

See Appendix B.

RSOS-210541.R1 (Revision)

Review form: Reviewer 1

Is the manuscript scientifically sound in its present form?

Yes

Are the interpretations and conclusions justified by the results?

Yes

Is the language acceptable?

Yes

Do you have any ethical concerns with this paper?

No

Have you any concerns about statistical analyses in this paper?

No

Recommendation?

Accept as is

Comments to the Author(s)

I believe that the authors have responded well to my suggestions and to those of the other reviewers. I have no additional suggestions or concerns.

Review form: Reviewer 3 (Kelly Dorgan)

Is the manuscript scientifically sound in its present form?

Yes

Are the interpretations and conclusions justified by the results?

Yes

Is the language acceptable?

Yes

Do you have any ethical concerns with this paper?

No

Have you any concerns about statistical analyses in this paper?

No

Recommendation?

Accept as is

Comments to the Author(s)

The authors did a really nice job of accurately and effectively addressing my comments and those of the other reviewers. The addition of predictions in the introduction made the manuscript much more cohesive. I really like the new figure 2 a lot – it is extremely effective at conveying the muscular differences between crawling and swimming lifestyles. This would be a great figure to include in a lecture or review paper to summarize the main findings of this paper. The topic sentences at the beginning of each methods section made the purpose of each section clear, and the addition of locomotory style to the musculature figures made the relevant differences much more apparent. Finally, the revised title much more clearly conveys the main findings of the article.

My one comment is a bit nit-picky but on page 12 line 33 where the authors refer to the fluid environment of the swimming worms as “low Reynolds number,” this should be changed to “intermediate Reynolds number”.

Aside from that minor comment, I felt that the manuscript was greatly improved and is an excellent contribution to our understanding of the functional morphology of polychaetes, and I recommend it for publishing.

Decision letter (RSOS-210541.R1)

Dear Dr Worsaae,

It is a pleasure to accept your manuscript entitled "Muscular adaptations in swimming scale worms (Polynoidae, Annelida)" in its current form for publication in Royal Society Open Science. The comments of the reviewer(s) who reviewed your manuscript are included at the foot of this letter.

on behalf of Dr Jonas Rubenson (Associate Editor) and Kevin Padian (Subject Editor)
openscience@royalsociety.org

Reviewer comments to Author:

Reviewer: 1

Comments to the Author(s)

I believe that the authors have responded well to my suggestions and to those of the other reviewers. I have no additional suggestions or concerns.

Reviewer: 3

Comments to the Author(s)

The authors did a really nice job of accurately and effectively addressing my comments and those of the other reviewers. The addition of predictions in the introduction made the manuscript much more cohesive. I really like the new figure 2 a lot – it is extremely effective at conveying the muscular differences between crawling and swimming lifestyles. This would be a great figure to include in a lecture or review paper to summarize the main findings of this paper. The topic sentences at the beginning of each methods section made the purpose of each section clear, and the addition of locomotory style to the musculature figures made the relevant differences much more apparent. Finally, the revised title much more clearly conveys the main findings of the article.

My one comment is a bit nit-picky but on page 12 line 33 where the authors refer to the fluid environment of the swimming worms as “low Reynolds number,” this should be changed to “intermediate Reynolds number”.

Aside from that minor comment, I felt that the manuscript was greatly improved and is an excellent contribution to our understanding of the functional morphology of polychaetes, and I recommend it for publishing.

Appendix A**ROYAL SOCIETY
OPEN SCIENCE****Parapodial myoanatomy reveals adaptations in swimming
Polynoidae (Aphroditiformia, Annelida)**

Journal:	Royal Society Open Science
Manuscript ID	RSOS-210541
Article Type:	Research
Date Submitted by the Author:	14-Apr-2021
Complete List of Authors:	Allentoft-Larsen, Marc; University of Copenhagen, Marine Biological Section Gonzalez, Brett C.; Smithsonian Institution, Smithsonian National Museum of Natural History, Department of Invertebrate Zoology Daniels, Joost; Monterey Bay Aquarium Research Institute Worsaae, Katrine; University of Copenhagen, Department of Biology Osborn, Karen; Smithsonian Institution, 4. Smithsonian National Museum of Natural History, Department of Invertebrate Zoology; University of California, Santa Cruz, Institute of Marine Sciences Katija, Kakani ; Monterey Bay Aquarium Research Institute
Subject:	behaviour < BIOLOGY, evolution < BIOLOGY, structural biology < BIOLOGY
Keywords:	musculature, scale worms, locomotion, 3D reconstruction, tomography, morphology
Subject Category:	Organismal and Evolutionary Biology

Author-supplied statements

Relevant information will appear here if provided.

Ethics

Does your article include research that required ethical approval or permits?:

This article does not present research with ethical considerations

Statement (if applicable):

The experiments in this study did not require an approval by an ethical committee. All procedures in this investigations comply with international and institutional guidelines, including the guidelines for animal welfare. Animals were collected under the scientific research permit (SRP # 18-12-01-23) issued by the TCI department of environmental and coastal resources (DECR), issued through TMI for sampling in Turks and Caicos, whereas collecting in other waters (e.g. Spain, Denmark) did not require sampling permits for segmented worms.

Data

It is a condition of publication that data, code and materials supporting your paper are made publicly available. Does your paper present new data?:

Yes

Statement (if applicable):

The datasets supporting this article are accessible via Dryad repository:

Allentoft-Larsen, Marc Christian et al. (2021), Gesiella jameensis MicroCT-scans for 3D reconstruction, Dryad, Dataset, <https://doi.org/10.5061/dryad.n02v6wwws>
<https://datadryad.org/stash/share/m8deTaZimGbnFXFiqMZXLj5AdDugWO5tGJWBCZ27aBw>

Allentoft-Larsen, Marc Christian et al. (2021), Macellicephalo longipalpa MicroCT-scans for 3d reconstruction, Dryad, Dataset, <https://doi.org/10.5061/dryad.ngf1vhhtg>
https://datadryad.org/stash/share/HDIlf0_RHObjhONFL4F6wFCYITV07w-JXxDyFqss63iM

Allentoft-Larsen, Marc Christian et al. (2021), Harmothoe imbricata MicroCT-scans for 3D reconstruction, Dryad, Dataset, <https://doi.org/10.5061/dryad.gqnk98smq>
https://datadryad.org/stash/share/woAuwqHG8diqMc_sLDbzbxMZht6hHXJatCua9Dze7ws

Allentoft-Larsen, Marc Christian et al. (2021), Pelagomacellicephalo iliffei MicroCT-Scans for 3D reconstruction, Dryad, Dataset, <https://doi.org/10.5061/dryad.4tmpg4f98>
https://datadryad.org/stash/share/Ur_snULWGty7fynhdefF-RL7XyDnA_K5G7tOftMeEMA

Allentoft-Larsen, Marc Christian et al. (2021), Branchipolynoe sp. MicroCT-Scans for 3D reconstruction, Dryad, Dataset, <https://doi.org/10.5061/dryad.kd51c5b5d>
https://datadryad.org/stash/share/PoNs6g51TIMsgp45nuKM6vCm5dpePtebdMCS3_njldc

Allentoft-Larsen, Marc Christian et al. (2021), EXTRA Drieschia sp. MicroCT-Scans for 3D reconstruction, Dryad, Dataset, <https://doi.org/10.5061/dryad.r2280gbcq>
https://datadryad.org/stash/share/ZPW8dq66-R9ExIxaRmcsqF_Ur_2YTSTB5-IgzkOpA54

Allentoft-Larsen, Marc Christian et al. (2021), Video recordings of polynoid (Annelida) swimming and
crawling, Dryad, Dataset, <https://doi.org/10.5061/dryad.0rxwdb0b>
https://datadryad.org/stash/share/pu0g3lZ9ROpT7PAvRJWFQXwLOINAAXLCKeCKzqxd_i4

***Conflict of interest***

I/We declare we have no competing interests

*Statement (if applicable):*

CUST_STATE_CONFLICT :No data available.

***Authors' contributions***

This paper has multiple authors and our individual contributions were as below

*Statement (if applicable):*

KW designed and coordinated the study. MCA, KW and BCG collected field data, carried out confocal
scanning microscopy along with morphological measurements, statistical analyses, 3D
reconstruction, video recordings, imaging and data analysis. BCG and KO generated microCT data.
MCA and KW drafted the
manuscript. MCA, BCG and KW participated in data analysis. JD carried out high-speed footage of
P. iliffei. MCA, KW, BCG, KO, KK and JD critically revised the manuscript. All authors gave final
approval for publication and agree to be held accountable for the work performed therein.

Parapodial myoanatomy reveals adaptations in swimming Polynoidae (Aphroditiformia, Annelida)

Marc C. Allentoft-Larsen¹, Brett C. Gonzalez², Joost Daniels³, Kakani Katija³, Karen Osborn^{2,3}, Katrine Worsaae^{1*}

1. Marine Biological Section, Department of Biology, University of Copenhagen, Universitetsparken 4, 2100-DK Copenhagen, Denmark.

2. Smithsonian National Museum of Natural History, P.O. Box 37012 Smithsonian Inst., 20013-7012, Washington D.C., USA

3. Monterey Bay Aquarium Research Institute, 7700 Sandholdt Road, Moss Landing, 95039, CA, USA

Keywords: musculature, scale worms, locomotion, 3D reconstruction, tomography

1. Summary

Annelids are predominantly found along the seafloor, but over time have colonised a vast diversity of habitats, such as the water column, where different modes of locomotion are necessary. Yet, little is known about their potential muscular adaptation to the continuously swimming required in the water column. The musculature and motility were examined for five scale worm species of Polynoidae (Aphroditiformia, Annelida) found in shallow waters, deep sea and caves that exhibit crawling, occasional swimming or continuous swimming, respectively. Their parapodial musculature was reconstructed using microCT and computational 3D analyses and the muscular functions interpreted from video recordings of their locomotion. Since most benthic annelids are able to swim for short distances using body and parapodial muscle movements, suitable musculature for swimming and a pelagic lifestyle is already present. Our results also indicate that rather than rearrangements or addition of muscles, a shift to a pelagic lifestyle is mainly accompanied by structural loss of muscle bundles and density, as well as elongation of extrinsic dorsal and ventral parapodial muscles. In addition, our study documents clear differences in locomotion and muscular arrangement among closely related annelids with different lifestyles as well as points to myoanatomical adaptations for accessing the water column.

*Authors for correspondence (kworsaae@bio.ku.dk)

†Present address: Marine Biological Section, Department of Biology, University of Copenhagen, Universitetsparken 4, 2100-DK Copenhagen, Denmark

2. Introduction

Annelida is a diverse animal group with about 21,000 extant species that have successfully colonised every marine environment, from benthic habitats in shallow waters to the deep-sea, and from the water column of the open ocean to that of caves (Fauchald, 1977; Weigert et al., 2016; Gonzalez et al., 2018; Gonzalez et al., 2020). In each environment annelids exhibit different lifestyles and locomotion, which are mediated by muscles of the body wall and parapodia if such are present (Tzetlin and Filippova, 2005; Kerbl et al., 2015; Struck et al., 2015). The body wall musculature always includes longitudinal muscles and a variation of either circular or transverse muscles and additional bracing muscles. Intermediate diagonal and oblique muscles may support locomotion and extend into the parapodia (Mettam, 1967; Storch, 1968; Tzetlin and Filippova, 2005). Members of the large clade Errantia are generally motile and errant with well-developed parapodia, the muscles of which typically include acicular, chaetal, ventral, and dorsal muscles, as well as parapodial wall muscles (Tzetlin and Filippova, 2005). However, the myoanatomy can vary greatly and there is no common muscular design for all annelids, nor within the two large and derived clades Sedentaria and Errantia (Mettam, 1967; Lawry Jr, 1971; Filippova et al., 2006; Müller and Worsaae, 2006).

Flexibility in the muscular system has made it possible to adapt locomotion to the different needs of a wide variety of habitats, enabling further shifts to new ecological niches (Westheide, 1997; Struck, 2011). Most benthic errantians may “occasionally swim” (Gray, 1939; Lawry Jr, 1971; Pettibone, 1986; Halanych et al., 2007), but cannot passively maintain their position within the water column. Benthopelagic species are primarily associated with the sea floor but depend on entering the water column above it to feed or reproduce. Some benthic species have a pelagic reproductive stage (epitoky) that transforms the structure and sometimes number of parapodia and chaetae, sensory organs, and musculature to swim into the water column to spawn (Clark, 1961; Bartels-Hardege and Zeeck, 1990; Pettibone, 1993). Exclusively or continuously swimming errantian annelids have originated independently within multiple families and are here defined as animals capable of spending long periods of time suspended in the water column, maintaining their position there. These “continuous swimmers” are found in the water column of the deep sea, open ocean and caves (Gonzalez et al., 2017), with those occurring in the open ocean referred to as holopelagic. The transition to life in the water column opens up a different set of prey and possible release from the high predation of the benthos and release from competition with benthic annelids (Rouse and Pleijel, 2001; Halanych et al., 2007). The transition to living exclusively in the water column has, in some cases, involved drastic morphological modifications such as transparent or gelatinous bodies and elongated appendages and parapodia (Halanych et al., 2007; Struck and Halanych, 2010; Osborn et al., 2011). The elongate parapodia are typically flexible and have elaborate oar-like structures (e.g.,

Tomopteridae; Gouveneaux, 2016). However, little is known about the evolution of specific morphological transformations facilitating their swimming lifestyle.

Morphological studies of benthic errantians have suggested that short and less manoeuvrable parapodia with numerous muscle groups such as those found in Aphroditidae, represent the plesiomorphic condition of recent Errantia (Tzetlin and Filippova, 2005). This muscle arrangement may be useful when dragging an elongate body across various substrates or even burrowing into them (Storch, 1968; Tzetlin and Filippova, 2005). Interestingly, reduction of muscle groups, bundles, fibres, and density may increase flexibility of the body and parapodia and has been suggested as an adaptation to life in the water column (Davenport and Bebbington, 1990; Tzetlin and Filippova, 2005; Filippova et al., 2006). Additionally, studies of pelagic and cave species have additionally suggested that elongated appendages may reduce sinking speed (Foxon, 1936; Davenport and Bebbington, 1990; Filippova et al., 2006; McNeill, 2015; Gonzalez et al., 2017) in addition to expanding the volume of water they can sense predators and prey in. Even though the musculature of annelids has been studied since the early 1900's, these studies mainly focused on *Aphrodite* and *Nereis* (Nereididae) (Fordham, 1925; Foxon, 1936; Gray, 1939), with few studies describing in detail the myoanatomy of parapodia and their part with locomotion. Comparative studies of parapodial myoanatomy are therefore important for understanding parapodial evolution and their functional significance for diversification of life styles (Mettam, 1967; Storch, 1968; Tzetlin and Filippova, 2005).

Scale worms (Aphroditiformia) are one of the most diverse annelid groups with more than 2000 described species (Zhang et al., 2018). Scale worms are characterised by the presence of scales (elytra) but exhibit highly divergent morphologies and lifestyles, inhabiting all known marine environments (Gonzalez et al., 2018; Zhang et al., 2018). They are largely epibenthic crawling predators with dorso-ventrally flattened bodies, well-defined parapodia and an eversible muscular pharynx with jaws (Fauchald, 1977; Gonzalez et al., 2018), such as seen in the shallow water polynoid *Harmothoe imbricata* (Lawry Jr, 1971; Zhang et al., 2017). However, continued midwater oceanic exploration has revealed an increased number of continuously swimming scale worms, highlighting the group's success in diversification and colonisation of new niches (Fauchald, 1977; Pettibone, 1985b, 1986; Struck and Halanych, 2010; Gonzalez et al., 2017). So far, only three truly holopelagic scale worm genera, i.e., *Drieschia*, *Drieschiopsis* and *Podarmus*, are described, and each is of unknown phylogenetic affinity (Støp-Bowitz, 1948; Dales and Peter, 1972; Støp-Bowitz, 1991). The deep-sea group Macellicephalinae (Polynoidae) is generally considered epibenthic, however, two monotypic genera known exclusively from anchialine caves, *Gesiella jameensis* and *Pelagomacellicephala iliffei* are found continuously swimming in the water column (Hartmann-Schröder, 1974; Pettibone, 1985a). Occasional swimming or swimming bouts have also been observed for other deep-sea scale worms, including the commensal *Branchipolynoe* within the closely related Lepidonotopodinae (Polynoidae). These branchiate polynoids inhabit deep-sea mussels, swimming between mussel patches

along the bottom. While they cannot passively maintain their position in the water column, they are effective swimmers (Pettibone, 1986; Hatch et al., 2020).

In order to unravel potential behavioural and morphological adaptations in swimming scale worms, their parapodial musculature were reconstructed using micro-computed tomography (microCT) scans and computational 3D muscle reconstructions and videos were recorded in five species of Polynoidae. These spanned four different lifestyles and habitats: i) benthic crawling, shallow water (*H. imbricata*, Polynoinae); ii) continuous swimming, anchialine cave water column (*G. jameensis*, *P. iliffei*, Macellicephalinae); iii) benthic crawling/possibly performing occasional swimming, deep-sea (*Macellicephalo longipalpa*, Macellicephalinae); and iv) occasional swimming, deep-sea commensal (*Branchipolynoe* sp., Lepidonotopodinae). Through comparison of muscle volume, muscle number, functionality and overall locomotion among the five examined species, we address whether 1) morphology and functionality of similar muscle groups differ between crawling and continuous swimming species, and 2) a continuous swimming is associated with behavioural and muscular adaptations. We identify myoanatomical adaptations necessary for accessing the water column and evolving continuous swimming.

3. Materials and Methods

3.1. Specimens and sampling

[revised manuscript text omitted]

4. Results

4.1. Parapodial muscles

Largely following the nomenclature of Mettam (1967 and 1971) and Lawry (1971), the parapodial musculature of the five studied polynoids comprises 20-21 muscle groups with 51-67 individual sub-bundles (Tables 1-3). Muscles are numbered consecutively #I-XXI and include complimentary anterior/posterior bundles (Table 2, Figs. 1-6): #I-II, dorsal muscles, #III-IV, ventral, #V-VI, oblique muscles, #VII-VIII, bracing muscles, #IX, dorso-ventral crossing muscle, #X, anterior-posterior transverse muscles, #XI, ventral diagonal muscles, #XII, notopodial acicular muscles, #XIII and XVII, chaetal protractor muscles, #XIV and XVIII, chaetal retractor muscles, #XV and XIX, chaetal sac muscles, #XVI, neuropodial acicular muscles, #XX, elyrophore muscles and #XXI, interacicular muscle. Table 2 lists the number of bundles and keywords for each of the 21 muscle groups for all species. Main characteristics of each muscle are given below, with detailed descriptions of individual muscle groups for each species listed in the Supplementary material 
In all species examined, the extrinsic dorsal muscles #I-II (presumed levator, #I, II, remotor, #II and promotor, #I muscles) and ventral muscles #III-IV (presumed remotor, #IV, promotor, #III and depressor, #III, IV muscles) constitute the most massive bundles, but numerous minor muscle groups are located intrinsically in the parapodium. The dorsal and ventral muscle groups #I-IV comprise complimentary anterior/posterior bundles, of which the anterior bundles are wider and thicker proximally (Figs. 2-3). The elyrophore is supported by a minor intrinsic, ventro-dorsally oriented parapodial muscle bundle #XX (depressor muscle) sheathed by supportive muscles (Fig. 1, #XX). The notopodium is slender and supplied by fewer muscle fibres than the neuropodium. It completely lacks extrinsic muscles and only has minor branches of fibres from the intrinsic chaetal muscles #XIII, #XIV, #XVI and muscles #I and #III (Figs. 2, S1-S4). The neuropodium is longer and broader and has both the extrinsic dorsal and ventral muscle groups (#I-IV) and the intrinsic muscles #XVII, #XVIII and #XIX (Fig. 1). The bracing muscles (#VII-#VIII) support the body wall between the parapodia (Figs. 2, 4). Oblique muscles (#V-VI) extend over the ventral muscles (#III-IV) from the ventral body midline to the body wall, likely supporting parapodial movements (Fig. 1). Inside the parapodium are numerous intrinsic muscle groups, a ventro-dorsal crossing muscle (#IX), numerous anterior-posterior transverse muscles, likely for keeping hydrostatic pressure (#X), and chaetal and acicular protraction and retraction muscles to move the chaetae and aciculae (#XII-XIX) (Figs. 1-2, S1-4, #IX, X, XII-XIX). No obvious circular muscles of the body wall were observed in any species. A very delicate outer net of thin, crisscrossing muscles surrounded the parapodial walls were observed by CLSM and phalloidin staining, possibly representing the parapodial wall musculature of Mettam (1971) (Fig. S5).

Gesiella jameensis has a relative parapodium length: body width ratio of 1.74 (measured at middle, broadest segment; Table 3). The parapodial musculature is comprised of 20 muscle groups with 51 individual muscle bundles, which are generally relatively thin and elongated (Table 2, Figs. 2-5). Interacicular muscle #XXI is lacking. Specifically, the dorsal and ventral muscles (#I-IV) are comprised of fewer slender muscle bundles compared to *H. imbricata*, *M. longipalpa* and *Branchiopolynoe* sp., however, its acicular and chaetal muscles (#XII-XIX) are more numerous (Figs. 2-5) For a detailed description, see Supplementary material.

Pelagomacellicephala iliffei has a relative parapodium length: body width ratio of 1.4 (measured at middle, broadest segment, Table 3). The parapodial musculature comprises 20 overall muscle groups with 52 individual muscle bundles (Table 2, Figs. 3-6; for myoanatomy overview see suppl. Mat. Fig. S1). The musculature of *P. iliffei* is relatively similar to that of *G. jameensis*, with elongated parapodia and slender dorsal and ventral muscles bundles (#I-IV, Figs. 3, S1). Interacicular muscle #XXI is lacking (Table 2, Fig. S1). The notopodial chaetal sac musculature (#XV) was difficult to trace and only one obvious notopodial chaetal protractor muscle #XIII was found (versus two in *G. jameensis*; Figs. 5, S1). *Pelagomacellicephala iliffei* also possesses three anterior and posterior oblique muscle bundles #V-VI, while *G. jameensis* only possesses one (Table 2, Fig. 3). The supportive muscle in the elytophore (#XX, Figs. 5, S1) is more prominent in *P. iliffei* than *G. jameensis*.

Macellicephala longipalpa has a relative parapodium length: body width ratio of 1.56 (measured at segment 5-9; Table 3). The parapodial musculature comprises 20 muscle groups with 51 individual muscle bundles, which are generally elongated and thin (Table 2, Figs. 3-6, S2). Interacicular muscle #XXI is lacking (Table 2, Fig. S2). The musculature of *M. longipalpa* generally follows that of both *G. jameensis* and *P. iliffei*, but detailed comparison of these three species shows that in *M. longipalpa* there are several slight differences as follows: 1) the parapodium is generally more slender, 2) the notopodium is relatively smaller, 3) bracing muscles are more prominent, 4) acicular muscles are fewer, 5) the intrinsic muscles (#IX,X, XII-XIX) are generally more tightly packed (Figs. 4, 5, S2), 6) the elytophore muscles are more numerous and more prominent and 7) the elytophore is located more distally on the parapodium (Figs. 2, 5, S2-S4) than seen in both *G. jameensis* and *P. iliffei*. For these comparisons, the age of the specimens should be considered because after 92 years in preservative, delicate tissue and muscles may have collapsed in the parapodium cavity, making the interior intrinsic musculature then also seem more tightly packed and shorter.

Harmothoe imbricata has a relative parapodium length: body width ratio of 0.81 (measured at middle, broadest segment, Table 3). The parapodial musculature comprises 21 muscle groups with 67 individual muscle bundles; interacicular muscle #XXI is present (Table 2, Fig. 6). The muscle groups are generally relatively short, dense, thick, tightly packed and consists of numerous muscle bundles; especially the acicular and chaetal muscles are dominant (Table 2, Figs. 3-6, S3). Ventral and dorsal muscles (#I-IV)

comprise more muscle bundles and sheet-like structures than found in the other species (Fig. 3). The numerous intrinsic muscle groups (#IX, X, XII-XIX, Figs. 4, 5) are tightly packed in the parapodium leaving little free space in the parapodial cavity. The neuropodium is twice the size of the notopodium with the ventral and dorsal muscle groups (#I-IV) extending extrinsically. The intrinsic notopodial musculature #XIII-XV is densely packed, but structurally follows that of the other species in number and thickness (Fig. 5). The elyrophore muscles #XX take up a large portion of the parapodium with thick and dense muscle bundles extending both intrinsically and extrinsically (Fig. 5).

Branchipolynoe sp. has a relative parapodium length: body width ratio of 1.12 (measured at middle, broadest segment, Table 3). The parapodial musculature comprises 20 muscle groups with 61 individual muscle bundles, which are generally relatively short and dense (Table 2, Figs. 3-6, S4). The muscular composition of *Branchipolynoe* sp. follows that of the above species and is most similar to *H. imbricata* when comparing muscle bundle numbers and density. The most noticeable differences from *H. imbricata* are the large separations between the branches of the extrinsic muscles #I-IV, where #IV distinguishes itself with distinct branching (Fig. 3). The relative length of neuropodium to notopodium is shorter than in the other species examined and the notopodium is shifted anteriorly relative to the neuropodium (Fig. S4). Anterior-posterior transverse muscles are more numerous (#X, Fig. 4). A large cavity between dorsal muscles and elyrophore muscles in the dorsal part of the parapodium gives space for extensive branchia (Fig. S4A), which are absent in the other species. The elyrophore muscles (#XX) are more numerous and distinct and the supportive dorsal muscle around it is prominent (Fig. 5). Noticeable is the great proportion of interparapodial bracing muscles (#VII, Fig. 4).

4.2. Parapodial muscle volumes

Muscle volumes were extracted from each parapodial reconstruction of the 21 larger muscle groups described above (Tables 2, 3). *Macellicephalo longipalpa* had the greatest total parapodial muscle volume (1.2026 mm³) and *P. iliffei* had the lowest (0.0347 mm³), despite differences in the number of muscle bundles and length of parapodia (Table 3). The dorsal and ventral muscles #I-IV dominate musculature by volume with similar relative volumes of 12.4-17.6%, except in *H. imbricata* that had both lower relative volumes (6.3%-9.5%) and no appreciable difference from those of oblique, acicular and elyrophore muscle groups (#V-VI, XII, XVI, XX). Bracing muscles (#VII-VIII) constituted high relative volumes in all species (up to 25.6% in *Branchipolynoe* sp.), except for the benthic crawling *H. imbricata* which had limited bracing muscles, constituting only 5.9% of the total volume. *Pelagomacellicephalo iliffei*, *G. jameensis*, *Branchipolynoe* sp. and *M. longipalpa* showed similar relative volumes to each other, especially in dorsal, ventral, anterior-posterior transverse and acicular/chaetal muscles, but differed in relative volumes of neuropodial chaetal sac and elyrophore muscles. *Harmothoe imbricata* was distinct from the others by having a more equal

[revised manuscript text omitted]

5. Discussion

Few papers have studied the parapodial muscles of annelids despite their importance in the evolution of locomotion and lifestyle. By comparing crawling, occasional swimming and continuous swimming, members of subfamilies Polynoinae, Macellicephalinae and Lepidonotopodinae in this study provides insight into the myoanatomical adaptations of locomotion in scale worms.

5.1. Muscle groups comparison between crawling and continuous swimming species

The parapodial myoanatomy of *G. jameensis*, *P. iliffei* and *M. longipalpa* are quite similar to each other in terms of number of bundles, structure, muscle proportions and point of origin/insertion. Despite the elongation and the reduced number of muscle bundles and mass, these species are also directly comparable to the myoanatomy found in *Branchipolynoe* sp. and *H. imbricata*. The arrangement of dorsal, ventral, oblique, bracing, elyrophore, acicular and chaetal muscles are directly comparable to the findings of this study and prior studies of scale worms and other errantian annelids (Mettam, 1967; Storch, 1968; Lawry Jr, 1971; Mettam, 1971; Weidhase et al., 2016). Even in Onychophorans, dorsal and ventral muscles (#I-IV) with points of origin and insertion much like that found in annelids are reported. Onychophorans even have intrinsic protractor and retractor muscles of the claw that are comparable to the chaetal muscles

in annelids (Oliveira et al., 2019). Despite these similarities, there are some differences in the detailed morphology of some muscle groups with regard to form, number and volume among the examined scale worms:

Dorsal and ventral muscles (#I-IV)

The dorsal anterior and posterior muscles (#I-II) and ventral anterior and posterior muscles (#III-IV) likely serve a key role in the oaring movements that create the power and recovery strokes (Mettam, 1967; Yang, 2012; Oliveira et al., 2019). Like prior studies, we found that in contrast to other studied errant annelids (e.g., *Nereis*), these muscles constitute most of the muscular volume in all scale worms studied thus far (Mettam, 1967; Storch, 1968; Mettam, 1971; Filippova et al., 2006; Weidhase et al., 2016).

Although the dorsal and ventral muscles (#I-IV) still dominate in relative volume, we found that these muscles groups are more elongated and thinner in the swimming species *G. jameensis*, *P. iliffei* and *M. longipalpa* (and presumably in *Drieschia* sp.) than in the predominantly crawling *H. imbricata* and *Branchipolynoe* sp. Myoanatomical studies of the occasional swimmers *Nereis* and *Nephtys* (Nephtyidae) likewise show elongated and thinner dorsal and ventral muscles, which are known to perform a similar function during locomotion (Clark and Clark, 1960; Mettam, 1967).

As such, we suggest this elongation of the dorsal and ventral muscles (#I-IV), and of parapodial muscles in general, to be an adaptation to a continuous swimming and necessary when the parapodia elongate, as seen in many swimming Errantia. Such lengthening has the biomechanical potential to make the parapodial drag more efficient for swimming locomotion (Yang, 2012; Takagi, 2015).

The somewhat greater volumes measured in most species of the anterior dorsal muscle (#I) and anterior ventral muscle (#III) compared to their posterior dorsal and ventral muscle counterpart (#II and #IV) (Table 4) is puzzling, since the thrust gaining power stroke during locomotion is posteriorly directed and denser posterior muscles presumably drive this movement. On the other hand, the recovery stroke is moving against the flow of water, thus, more strength may be required from the anterior muscle to counter this additional drag. However, as there is no space between the parapodia during the recovery stroke, the exposure to the fluid is limited and minimizes drag (Yang, 2012).

In previous papers, the ventral muscles (#III-IV) have been divided into additional muscle groups, including the muscle ‘obliquus ventralis’ (Storch 1968), later changed to ‘parapodial oblique’ muscle by Mettam (1971) and the ‘ventral parapodial muscles’, which are called ‘lattice structures’ in the neuropodium by Mettam (1971). Previous investigations have relied on histology, transmission- and electron scanning microscopy, in which important structures and 3D orientations may be misinterpreted

(Tzetlin and Filippova, 2005). The microCT scanning gives a much better volume and 3D overview of the muscle structures. However, microCT scanning may exclude some delicate tissue layers, i.e., mesentery, peritoneum and septum, and bundles that are intertwined and cross closely can be difficult to resolve. Because of the difficulty seeing these very fine details, it is possible that the ventral muscles (#II-IV) in this study are also divided into oblique ventral parapodial muscles and lattice ventral parapodial wall muscles.

Acicular and chaetal muscles (#XII-XIX)

The acicular and chaetal muscles (#XII-XIX) supporting the chaetae are likely associated with locomotion and protection (deter predation by splaying their chaetae) and are well-developed in the examined benthic crawling species (Lawry Jr, 1971; Mettam, 1971; Tzetlin and Filippova, 2005). In *H. imbricata* these muscles take up 26.9% (compared to 9.4-15.9% in the other examined species) of the total parapodial muscle volume. Likely, the stout and numerous chaetae in *H. imbricata* need further muscular support for extension, retraction, elevating and depression when crawling through mud and sand (Mettam, 1971; Filippova et al., 2006; Müller and Worsaae, 2006). In contrast, swimming species would only need to control the spread of the chaetae to optimize for maximum drag during power and minimum drag during recovery strokes and possibly to stabilize or stiffen them during power strokes.

Interacicular muscle (#XXI)

The interacicular muscle (#XXI) connects the proximal ends of the noto- and neuroacicula. While few papers address the function of the interacicular muscle, both Lawry (1971) and Mettam (1971) indicate that they help the parapodium resist the drag and pull involved in crawling or digging through sediment by functioning as an antagonist in the movement of the aciculae. In pelagic and continuous swimming annelids, the parapodium rarely comes in contact with substrate and therefore does not need the muscular support of the aciculae. Interestingly, the interacicular muscle (#XXI) is lacking in the highly mobile and swimming species examined (*G. jameensis* and *P. iliffei*) as well as in the incomplete reconstructions of *Drieschia* sp. Interestingly, interacicular muscles are also lacking in the genus *Nephtys*, but earlier parapodial investigations observed them in other benthic, crawling species of Aphroditidae, Polynoidae, Sigalionidae and Nereididae (Storch, 1968; Tzetlin and Filippova, 2005). The holopelagic *Tomopteris* lacks aciculae completely, likely to enhance parapodial flexibility even further. Extra muscles associated with the aciculae may even decrease flexibility and movement of the parapodium (Filippova et al., 2006). In Sedentaria families like Terebellidae and Scalibregmidae, these muscles are further connected with muscle fibres that originate at the ventral midline (Storch, 1968; Tzetlin and Filippova, 2005; Filippova et al., 2006), presumably aiding to the anchoring of their chaetae in tubes and burrows.

Bracing muscles (#VII-VIII)

It has been suggested that the lack of circular muscles in many errant annelids is compensated for by

bracing muscles (Tzetlin et al., 2002; Tzetlin and Filippova, 2005; Filippova et al., 2006). The bracing muscles are normally associated with maintaining the body wall constrictions and peristaltic movement (Tzetlin and Filippova, 2005). Bracing muscles vary greatly among our studied scale worms. They are thinner in *G. jameensis* and *P. iliffei*, whereas in *H. imbricata*, they are dense and thick. In *Branchipolynoe* sp. these muscles have the greatest observed proportions and take up 25% of the total muscle volume (Table 4). For *Aphrodita aculeata*, the bracing muscles are more complex and comprise three rows of dorsal, lateral and ventral muscles (Mettam, 1971), likely supporting the additional parapodial strength required when burrowing and moving in the sediment. In other annelids, the bracing muscles are even more diverse and varied, with large lateral crossed bundles in *Nereis*, to thin dorsal fibres in *Dorvillea* (Dorvilleidae) (Mettam, 1967; Tzetlin and Filippova, 2005; Filippova et al., 2006). The greater proportion of bracing muscles likely support crawling and aid in twisting, turning and constriction of the body. Given the broad (or well-developed) bracing muscles in *Branchipolynoe*, we interpret this as an adaptation for manoeuvring in and out of mussels and around rugged mussel beds.

Elytrophore muscles (#XX)

These muscles are only found in scale worms since they are the only annelids to possess elytra. However, this musculature may be homologous to that of the cirrophore of the dorsal cirrus, a structure found in many other annelids. Both cirrophores and elytrophores comprise extrinsic and intrinsic muscles to depress and levitate either the elytra or the sensing style of the dorsal cirrus (Mettam, 1971). The elytrophore muscle bundles are both more numerous and denser in the benthic *H. imbricata*, having six bundles (Fig. 5), than in the continuous swimming cave species *G. jameensis* and *P. iliffei* only possessing an elytrophore depressor muscle and a supportive elytrophore wall musculature. These muscular reductions in the cave species likely reflect the otherwise protective and ventilating roles of the elytra in open water, benthic species. *Harmothoe imbricata*, living in shallow water habitats, is capable of illuminating its elytra upon disturbance and even shed them when attacked to further distract its decapod and fish predators (Lawry Jr, 1971; Miron et al., 1987). In caves where larger predators are either entirely lacking or visually limited, scale worms like *G. jameensis* and *P. iliffei* are among the top predators (Martínez, 2016; Gonzalez et al., 2020). Therefore, defensive movements and shedding of the elytra may be less important and hence, less muscles are allocated to the elytrophore. The elytra of these cave scale worms are also thinner and lighter than other scale worm elytra, which would require less muscles and their reduction possibly further aid to an increased buoyancy of these swimming species (Pettibone, 1985b). *Harmothoe imbricata* also lives in the captured sediment of mussel beds with decaying mussels and possibly low oxygen conditions where elytra-guided ventilation may be beneficial. The benthopelagic *Branchipolynoe* sp. also lives and moves between and within mussels with limited water flow but has extended branchia for oxygen uptake and smaller elytra (Pettibone, 1986). However, even though their elytra are highly reduced *Branchipolynoe* sp.

has an intermediate number of four elytophore muscles, which may therefore be a relic from a more crawling ancestor and/ or related to ventilation. 
5.2. Behavioural and muscular adaptations to a continuous swimming lifestyle in Annelids

Our video recordings show that all examined species were able to swim by the use of parapodial power and recovery strokes in combination with body undulations (Fig. 7). Swimming may be even more prevalent in scale worms than previously thought based on the number of in situ observations (Pettibone, 1986, 1993; Parzefall, 1996; Gonzalez et al., 2017). However, when examining their swimming behaviour in detail, the degree of swimming ability varies greatly. Even though *H. imbricata* can swim, it only hovers a few millimetres off the substrate for short distances, and primarily moves by crawling. Videos of *Lepidonotus*, another shallow water scale worm, show similar swimming just above the sediment (Kreutzer et al., 1989). In the opposite extreme, both *G. jameensis* and *P. iliffei* move exclusively by swimming and drifting in the water column. Their drifting is likely aided by the observed outstretching of their dorsal cirri, reducing their sinking rate by potentially increasing drag (Fig. 8A). However, during collection of *G. jameensis*, we observed that their escape response was to head into cave cracks and crevices, which required crawling. *Gesiella jameensis* and *P. iliffei* both have excellent swimming abilities and show similar parapodial movements to the holopelagic *Tomopteris* (Halanych et al., 2007; Aoki et al., 2016; Gonzalez et al., 2017; Daniels et al., 2021).

The elongated parapodia observed in *G. jameensis* and *P. iliffei* are typical of many members of the deep-sea Macellicephalinae, which are assumed to be epibenthic and capable of swimming but have yet to be observed doing so. Similarly, elongated parapodia are widespread across holopelagic annelids, including all members of Tomopteridae, Alciopini and Lopadorrhynchidae. These parapodia evidently function as 'oars' and their surface area provides greater thrust and possibly helps achieve greater velocity and efficiency (Takagi, 2015). The proposed myoanatomical features associated with continuous swimming, elongated parapodia muscles and fewer and thinner muscle bundles were observed in *M. longipalpa* (Macellicephalinae). This suggest that *M. longipalpa* is a capable swimmer, however, its total and relative parapodial muscle volume likely prevent it from staying in the water column when not actively swimming (Pettibone, 1976; Pettibone, 1985a). *Branchipolynoe* sp. exhibits a musculature that shares elongated extrinsic dorsal and ventral muscles and thin acicular chaetal muscles with *G. jameensis* and *P. iliffei*, and numerous dense musculature with *H. imbricata*, which fits well with its reported benthopelagic lifestyle and periodic swimming between bivalve hosts (Pettibone, 1986).

Crawling does not require long parapodia since it relies on contact with the substrate and movements of strong depressor, retractor and protractor muscles for lift and forward motion. In a benthic environment, it

is thus likely that short parapodia with numerous, dense muscle bundles and high relative muscle volume, as documented here for *H. imbricata* and *Branchipolynoe* sp., are beneficial when crawling (Lawry Jr, 1971; Kirkegaard, 1992; Yang, 2012; Oliveira et al., 2019). A similar muscle arrangement is also found in other benthic crawling and digging scale worms i.e., *Lepidonotus squamatus* and *A. aculeata*, the latter possessing even more complex and compact musculature (Storch, 1968; Mettam, 1971). Even though data on parapodial musculature in sedentarian annelids are scarce, the few studies available suggest well-muscled septa and extra acicular muscle connections to the ventral midline are important for enhanced efficiency in burrowing (Storch, 1968; Tzetlin and Filippova, 2005).

On the other hand, increased buoyancy, swimming ability and manoeuvrability in the water column is important in pelagic and continuous swimming lifestyles. Swimming efficiency is likely facilitated by elongated parapodia and the accompanying elongation of parapodial musculature. Greater parapodial length will also allow more efficient displacement of the surrounding fluids and increased locomotion (Bhaud and Cazaux, 1990; Davenport and Bebbington, 1990; McNeill, 2015; Gonzalez et al., 2017). Elongated appendages and their posture when drifting, such as the observed outstretched dorsal cirri in the two cave worm species, increase surface to volume ratio and thus slow sinking. Likely more important than increasing the surface to volume ratio for extended time in the water column, is the reduced muscle number, density and relative volume documented here for the cave worms *G. jameensis* and *P. iliffei* and indicated in the holopelagic *Drieschia*. Buoyancy will generally increase with reduced muscle weight and more space for body fluid that is more similar in density to seawater than muscle tissue. This is best observed in Tomopteridae where their myoanatomy has been reduced to a thin mesh just below the epidermis, leaving a large fluid-filled body cavity (Meyer, 1929). In addition to reduced weight, loss of some muscle groups (e.g., the interacicular muscles as documented here; (Lawry Jr, 1971; Mettam, 1971) may increase parapodial flexibility. Increased parapodial flexibility may be further supported by development of other muscle groups such as elongated parapodial musculature (Storch, 1968; Tzetlin and Filippova, 2005; Müller and Worsaae, 2006; Purschke and Müller, 2006).

6. Conclusion

This study has provided insight into the myoanatomy associated with locomotion in scale worms. By comparing different scale worms and lifestyles, common muscle groups were identified, and their function predicted. Differences were also detected through detailed comparison of the parapodial myoanatomy, which allowed identification of myoanatomical adaptations to crawling and swimming, respectively. Adaptations for continuous swimming are a reduction in the number, density, volume and lengthening of muscles, as seen in *G. jameensis* and *P. iliffei*. Both the elongated parapodia and dorsal cirri that provide a

larger surface area to volume ratio and the fewer, thinner muscles that decrease body mass and density, increase buoyancy and their ability to passively maintain their position in the water column during swimming and drifting. These results showed that even though polynoids exhibit relatively similar muscle arrangements in terms of point of origin and insertion, the locomotion in crawling and swimming species is mediated by different muscle groups. The dorsal and ventral muscles #I-IV likely serve a key role in swimming and oaring movements of the parapodia, and elongation of these muscles is necessary for longer parapodia. Furthermore, in crawling annelids, well-developed acicular and chaetal muscles are also important when moving through sediment (Mettam, 1971; Filippova et al., 2006). The interacicular muscle is not present in species with continuous swimming abilities and is mainly there to support the acicular position during crawling locomotion.

The overall muscle arrangement is similar in swimming and crawling scale worms but inhabiting the water column has resulted in modifications to muscle volume, shape and number. However, small swimming forays are still common and possible in benthic scale worms using parapodial movements and body undulations supported by strong parapodial and longitudinal muscles. Detailed locomotion descriptions for both crawling and continuous swimming species were presented above. Our locomotion analyses reveal that, even though they are not sustained for longer periods of time, benthic crawling species have similar swimming patterns to continuously swimming species, even.

Through morphological and behavioural comparison between different scale worms and their locomotion, we have gained great insight into the morphological changes associated with accessing the water column. However, continuous swimming species were only represented here by cave-dwelling species. Our preliminary study of *Drieschia*, although the material was very limited, suggested that more extreme cases of swimming annelids (e.g., *Drieschia* and tomopterids) may have undergone even further reduction of parapodial muscles. Morphological studies of these holopelagic annelids would help us understand the extremes of the scale worm swimming adaptations, provide a unique lineage for comparison, and better document the adaptive nature of annelids as a whole.

Acknowledgements

We would like to thank the members of the 'Caicos Caves III' team during the 2019 expedition to Turks and Caicos: Thomas Iliffe, Jørgen Olesen, Sarit B. Truskey, Lauren Ballou, Paul Heinerth, Mark Parrish, Naqqi Manco and Jon Ward. For collection of *Gesiella* we thank the local Environmental Service of the Cabildo de Lanzarote, Alejandro Garcia Martínez, Malte Jarlgaard Hansen and Alvaro Garcia Herrero, all aiding with permissions and logistics to access Túnel de la Atlántida and Cueva De Los Lagos for diving and collecting of

samples. Thanks to Paula Mendoza (student of Katrine Worsaae) for the collection of *H. Imbricata*. We are
grateful to Sofie Gudnitz-Larsen for her help and support with illustrations and to Freya Goetz for here
assistance with the microCT scanning. Special thanks to Alexandra Kerbl and María Herranz for their initial
support and discussion. Lastly, Sanni Jensen deserves a great thanks for her guidance and support of the
study process.

**Ethical Statement**

All applicable international, national, and/or institutional guidelines for animal testing, animal care and use of
animals were followed by the authors.

**Data Accessibility Statement**

The datasets supporting this article are accessible via Dryad repository:

Allentoft-Larsen, Marc Christian et al. (2021), *Gesiella jameensis* MicroCT-scans for 3D reconstruction, Dryad,
Dataset, <https://doi.org/10.5061/dryad.n02v6wwws>
<https://datadryad.org/stash/share/m8deTaZimGbnFXFiqMZXLj5AdDugWO5tGJWBCZ27aBw>

Allentoft-Larsen, Marc Christian et al. (2021), *Macellicephala longipalpa* MicroCT-scans for 3d reconstruction,
Dryad, Dataset, <https://doi.org/10.5061/dryad.ngf1vhhtg>
https://datadryad.org/stash/share/HDIlf0_RHObjhONFL4F6wFCYITV07w-JXxDyFqss63iM

Allentoft-Larsen, Marc Christian et al. (2021), *Harmothoe imbricata* MicroCT-scans for 3D reconstruction,
Dryad, Dataset, <https://doi.org/10.5061/dryad.gqnk98smq>
https://datadryad.org/stash/share/woAuwqHG8diqMc_sLDbzbxMZHT6hHXJatCua9Dze7ws

Allentoft-Larsen, Marc Christian et al. (2021), *Pelagomacellicephala iliffei* MicroCT-Scans for 3D
reconstruction, Dryad, Dataset, <https://doi.org/10.5061/dryad.4ttmpg4f98>
https://datadryad.org/stash/share/Ur_snULWGty7fynhdefF-RL7XyDnA_K5G7tOftMeEMA

Allentoft-Larsen, Marc Christian et al. (2021), *Branchipolynoe* sp. MicroCT-Scans for 3D reconstruction,
Dryad, Dataset, <https://doi.org/10.5061/dryad.kd51c5b5d>
https://datadryad.org/stash/share/PoNs6g51TIMsgp45nuKM6vCm5dpePtebdMCS3_njldc

Allentoft-Larsen, Marc Christian et al. (2021), *Drieschia* sp. MicroCT-Scans for 3D reconstruction, Dryad,
Dataset, <https://doi.org/10.5061/dryad.r2280gbcq>
https://datadryad.org/stash/share/ZPW8dq66-R9ExIxaRmcsqF_Ur_2YTSTB5-lgzkOpA54

Allentoft-Larsen, Marc Christian et al. (2021), Video recordings of polynoid (Annelida) swimming and
crawling, Dryad, Dataset, <https://doi.org/10.5061/dryad.0rxwdb50b>
https://datadryad.org/stash/share/pu0g3lZ9ROpT7PAvRjWfQXwLOINAAXLcEcKzqxd_i4

**Funding Statement**

Field and laboratory work during the expedition to Turks and Caicos in 2019 was financially supported by
The Smithsonian's Global Genome Initiative (GGI-2019-Rolling-214), the Danish Natural History Society and

the Peter Buck Fellowship Program. Further laboratory work was supported by the Carlsberg Foundation (grant # CF15-0946) and the David and Lucile Packard Foundation.

Authors' Contributions

KW designed and coordinated the study. MCA, KW and BCG collected field data, carried out confocal scanning microscopy along with morphological measurements, statistical analyses, 3D reconstruction, video recordings, imaging and data analysis. BCG and KO generated microCT data. MCA and KW drafted the manuscript. MCA, BCG and KW participated in data analysis. JD carried out high-speed footage of *P. iliffei*. MCA, KW, BCG, KO, KK and JD critically revised the manuscript. All authors gave final approval for publication and agree to be held accountable for the work performed therein.

References

- Allentoft-Larsen, M. C., B. C. Gonzalez, J. Daniels, K. Katija, K. Osborn, and K. Worsaae. 2021. Branchipolynoe sp. MicroCT-scans for 3D reconstruction. Dryad. Dataset. doi: [10.5061/dryad.kd51c5b5d](https://doi.org/10.5061/dryad.kd51c5b5d)
- Allentoft-Larsen, M. C., B. C. Gonzalez, J. Daniels, K. Katija, K. Osborn, and K. Worsaae. 2021. Drieschia sp. MicroCT-scans for 3D reconstruction. Dryad. Dataset. doi: [10.5061/dryad.r2280gbcq](https://doi.org/10.5061/dryad.r2280gbcq)
- Allentoft-Larsen, M. C., B. C. Gonzalez, J. Daniels, K. Katija, K. Osborn, and K. Worsaae. 2021. Gesiella jameensis MicroCT-scans for 3D reconstruction. Dryad. Dataset. doi: [10.5061/dryad.n02v6wws](https://doi.org/10.5061/dryad.n02v6wws)
- Allentoft-Larsen, M. C., B. C. Gonzalez, J. Daniels, K. Katija, K. Osborn, and K. Worsaae. 2021. Harmothoe imbricata MicroCT-scans for 3D reconstruction. Dryad. Dataset. doi: [10.5061/dryad.gqnk98smq](https://doi.org/10.5061/dryad.gqnk98smq)
- Allentoft-Larsen, M. C., B. C. Gonzalez, J. Daniels, K. Katija, K. Osborn, and K. Worsaae. 2021. Macellicephala longipalpa MicroCT-scans for 3D reconstruction. Dryad. Dataset. doi: [10.5061/dryad.ngf1vhtg](https://doi.org/10.5061/dryad.ngf1vhtg)
- Allentoft-Larsen, M. C., B. C. Gonzalez, J. Daniels, K. Katija, K. Osborn, and K. Worsaae. 2021. Pelagomacellicephala iliffei MicroCT-scans for 3D reconstruction. Dataset. doi: [10.5061/dryad.4tmpg4f98](https://doi.org/10.5061/dryad.4tmpg4f98)
- Allentoft-Larsen, M. C., B. C. Gonzalez, J. Daniels, K. Katija, K. Osborn, and K. Worsaae. 2021. Video recordings of polynoid (Annelida) swimming and crawling. Dryad. Dataset. doi: [10.5061/dryad.0rxwdbs0b](https://doi.org/10.5061/dryad.0rxwdbs0b)
- Aoki, N., N. Mushegian, K. Katija, and K.

- Osborn. 2016. A kinematic description of locomotion in the marine polychaete genus *Tomopteris*, Poster presentation. <https://naturalhistory.si.edu/sites/default/files/media/file/2016-aoki-poster.pdf>.
- Barnich, R., D. Fiege, G. Micaletto, and M. Gambi. 2006. Redescription of *Harmothoe spinosa* Kinberg, (Polychaeta: Polynoidae) and related species from Subantarctic and Antarctic waters, with the erection of a new genus. *Journal of Natural History* 40(1-2):33-75.
- Bartels-Hardege, H. D., and E. Zeeck. 1990. Reproductive behaviour of *Nereis diversicolor* (Annelida: Polychaeta). *Marine Biology* 106(3):409-412. doi: 10.1007/BF01344320
- Bhaud, M. R., and C. P. Cazaux. 1990. Buoyancy characteristics of *Lanice conchilega* (Pallas) larvae (Terebellidae). Implications for settlement. *Journal of Experimental Marine Biology and Ecology* 141(1):31-45. doi: 10.1016/0022-0981(90)90155-6
- Chernyshev, A. V. 2015. CLSM Analysis of the Phalloidin-Stained Muscle System of the Nemertean Proboscis and Rhynchocoel. *Zoological Science* 32(6):547-560. doi: 10.2108/zs140267
- Clark, R. 1961. The origin and formation of heteronemertis. *Biological Reviews* 36(2):199-236.
- Clark, R. B., and M. E. Clark. 1960. The Ligamentary System and the Segmental Musculature of *Nephtys*. *Quarterly Journal of Microscopical Science* 53(101(54)):149.
- Dales, R. P., and G. Peter. 1972. A synopsis of the pelagic Polychaeta. *Journal of Natural History* 6(1):55-92.
- Daniels, J., N. Aoki, J. Havassy, K. Katija, and K. J. Osborn. 2021. Metachronal swimming with flexible legs: A kinematics analysis of the midwater polychaete *Tomopteris*. Submitted March 2021.
- Davenport, J., and A. Bebbington. 1990. Observations on the swimming and buoyancy of some thecosomatous pteropod gastropods. *Journal of Molluscan Studies* 56(4):487-497. doi: 10.1093/mollus/56.4.487
- Fauchald, K. 1977. The polychaete worms. Definitions and keys to the orders, families and genera. *Natural History Museum of Los Angeles County, Science Series*
- Filippova, A., G. Purschke, A. B. Tzetlin, and M. C. M. Müller. 2006. Three-dimensional reconstruction of the F-actin musculature of *Dorvillea kastjani* (Dorvilleidae, Polychaeta) by means of phalloidin-labelling and cLSM. *Scientia Marina* 70(S3):293-300. doi: 10.3989/scimar.2006.70s3293
- Fordham, M. G. 1925. *Aphrodite aculeata*. University Press of Liverpool.
- Foxon, G. E. H. 1936. XL.— Observations on the locomotion of some Arthropods and annelids. *Annals and Magazine of Natural History* 18(106):403-419. doi: 10.1080/00222933608655210
- Gonzalez, B. C., A. Martínez, E. Borda, T. M. Iliffe, D. Eiby-Jacobsen, and K. Worsaae. 2018. Phylogeny and systematics of Aphroditiformia. *Cladistics* 34(3):225-259. doi: 10.1111/cla.12202
- Gonzalez, B. C., A. Martínez, J. Olesen, S. B. Truskey, L. Ballou, M. C. Allentoft-Larsen, J. Daniels, P. Heinerth, M. Parrish, and N. Manco. 2020.

Anchialine
biodiversity in the
Turks and Caicos
Islands: New
discoveries and
current faunal
composition.
International Journal
of Speleology 49(2):1.
Gonzalez, B. C., K. Worsaae,
D. Fontaneto, and A.
Martínez. 2017.
Anophthalmia and
elongation of body
appendages in cave
scale worms
(Annelida:
Aphroditiformia).
Zoologica Scripta
47(1):106-121. doi:
10.1111/zsc.12258
Gray, J. 1939. Studies in
Animal Locomotion.
Journal of
Experimental Biology
16(1):9.
Halanych, K. M., L. N. Cox,
and T. H. Struck.
2007. A brief review
of holopelagic
annelids. Integrative
and Comparative
Biology 47(6):872-
879. doi:
10.1093/icb/icm086
Hartmann-Schröder, G. 1974.
Die Unterfamilie
Macellicephalinae H.-
S.(1971)(Polynoidae,
Polychaeta). Mit
Beschreibung einer
neuen Art, M.
jameensis nov. spec.,
aus einem Gewässer
von Lanzarote (Kan.
Inseln). Mitt Hamb
Zool Mus Inst 71:23-
33.
Hatch, A. S., H. Liew, S.
Hourdez, and G. W.
Rouse. 2020. Hungry
scale worms:
- Phylogenetics of
Peinaleopolynoe
(Polynoidae,
Annelida), with four
new species. Zookeys
932:27-74. doi:
10.3897/zookeys.932.
48532
Hedrick, T. L. 2008.
Software techniques
for two- and three-
dimensional
kinematic
measurements of
biological and
biomimetic systems.
Bioinspiration &
Biomimetics
3(3):034001. doi:
10.1088/1748-
3182/3/3/034001
Kerbl, A., N. Bekkouche, W.
Sterrer, and K.
Worsaae. 2015.
Detailed
reconstruction of the
nervous and muscular
system of
Lobatocerebridae with
an evaluation of its
annelid affinity. BMC
Evolutionary Biology
15(1):277. doi:
10.1186/s12862-015-
0531-x
Kirkegaard, J. 1992.
Havborsteorme 1,
Errantia. Danmarks
Fauna 83:1-416.
Kreutzer, U., B. R.
Siegmond, and M. K.
Grieshaber. 1989.
Parameters
controlling opine
formation during
muscular activity and
environmental
hypoxia. Journal of
Comparative
Physiology B
159(5):617-628.
- Lawry Jr, J. V. 1971. The
parapodial and
segmental
musculature of
Harmothoë imbricata
(L.). Journal of
Morphology
135(3):259-272. doi:
10.1002/jmor.105135
0302
Martínez, A. 2016. Guide to
the anchialine
ecosystems of Los
Jameos del Agua and
Túnel de la Atlántida.
Cabildo de Lanzarote.
McNeill, A., R. . 2015. Size,
Speed and Buoyancy
Adaptations in
Aquatic Animals1.
American Zoologist
30(1):189-196. doi:
10.1093/icb/30.1.189
Mettam, C. 1967. Segmental
musculature and
parapodial movement
of *Nereis diversicolor*
and *Nephtys*
homborgi (Annelida:
Polychaeta). Journal
of Zoology
153(2):245-275. doi:
10.1111/j.1469-
7998.1967.tb04062.x
Mettam, C. 1971. Functional
design and evolution
of the polychaete
Aphrodite aculeata.
Journal of Zoology
163(4):489-514. doi:
10.1111/j.1469-
7998.1971.tb04546.x
Meyer, A. 1929. On the
coelomic cilia and
circulation of the
body fluid in
Tomopteris
helgolandica. Journal
of the Marine
Biological
Association of the

United Kingdom 10.1111/j.1096- Pettibone, M. H. 1986. A new
16(1):271-276. 3642.2011.00727.x scale-worm
Miron, M.-J., L. LaRivière, Parzefall, J. 1996. commensal with
7 J.-M. Bassot, and M. Behavioural and deep-sea mussels in
Anctil. 1987. morphological changes caused by the seep-sites at the
Immunohistochemical light conditions in Florida Escarpment in
and radioautographic deep-sea and shallow- the eastern Gulf of
evidence of water habitats. na. Mexico (Polychaeta:
monoamine- Polynoidae:
containing cells in Branchipolynoinae).
bioluminescent elytra Revision of the genus Proceedings of the
of the scale-worm Macellicephala Biological Society of
*Harmothoe imbricata* McIntosh and the Washington
(Polychaeta). Cell and subfamily
Tissue Research Macellicephalinae
249(3):547-556. doi: Hartmann-SchrAder
10.1007/BF00217326 (Polychaeta:
Müller, M. C. M., and K. Polynoidae).
Worsaae. 2006. Smithsonian
CLSM analysis of the Contributions to
phalloidin-stained Zoology
muscle system in Pettibone, M. H. 1985a. New
*Nerilla antennata*, genera and species of
*Nerillidium* sp. and deep-sea
*Trochonerilla mobilis* Macellicephalinae and
(Polychaeta; Harmothoinae
*Nerillidae*). Journal of (Polychaeta:
Morphology Polynoidae) from the
267(8):885-896. doi: hydrothermal rift
10.1002/jmor.10292 areas off the
Oliveira, I. d. S., A. Galapagos and
Kumerics, H. Jahn, Western Mexico at 21
36 M. Müller, F. Pfeiffer, °N and from the Santa
and G. Mayer. 2019. Catalina Channel.
Functional Proceedings of the
morphology of a Biological Society of
lobopod: case study of Washington
an onychophoran leg. 98(3):740-757.
Royal Society Open Pettibone, M. H. 1985b.
Science Polychaete worms
6(10):191200. doi: from a cave in the
10.1098/rsos.191200 Bahamas and from
Osborn, K. J., S. H. D. experimental wood
Haddock, and G. W. panels in deep water
Rouse. 2011. *Swima* of the North Atlantic
(Annelida, Polynoidae:
Acrocirridae), Macellicephalinae,
holopelagic worms Harmothoinae).
from the deep Pacific. Proceedings of the
Zoological Journal of Biological Society of
the Linnean Society Washington
163(3):663-678. doi: 98(1):127-149.
Pettibone, M. H. 1976. Revision of the genus
Macellicephala
McIntosh and the
subfamily
Macellicephalinae
Hartmann-SchrAder
(Polychaeta:
Polynoidae).
Smithsonian
Contributions to
Zoology
Pettibone, M. H. 1985a. New
genera and species of
deep-sea
Macellicephalinae and
Harmothoinae
(Polychaeta:
Polynoidae) from the
hydrothermal rift
areas off the
Galapagos and
Western Mexico at 21
°N and from the Santa
Catalina Channel.
Proceedings of the
Biological Society of
Washington
98(3):740-757.
Pettibone, M. H. 1985b.
Polychaete worms
from a cave in the
Bahamas and from
experimental wood
panels in deep water
of the North Atlantic
(Polynoidae:
Macellicephalinae,
Harmothoinae).
Proceedings of the
Biological Society of
Washington
98(1):127-149.
Pettibone, M. H. 1993.
Revision of some
species referred to
Antinoe, *Antinoella*,
Antinoana, *Bylgides*,
and *Harmothoe*
(Polychaeta:
Polynoidae:
Harmothoinae).
Smithsonian
contributions to
zoology
Purschke, G., and M. C. M.
Müller. 2006.
Evolution of body
wall musculature.
Integrative and
Comparative Biology
46(4):497-507. doi:
10.1093/icb/icj053
Rouse, G., and F. Pleijel.
2001. Polychaetes.
Oxford university
press.
Storch, V. 1968. Zur
vergleichenden
anatomie der
segmentalen
muskelsysteme und
zur verwandtschaft
der polychaeten-
familien. Zeitschrift
für Morphologie der
Tiere 63(3):251-342.
doi:
10.1007/BF00292073
Struck, T. H. 2011. Direction
of evolution within
Annelida and the

- definition of Pleistoannelida. *Journal of Zoological Systematics and Evolutionary Research* 49(4):340-345. doi: 10.1111/j.1439-0469.2011.00640.x
- Struck, Torsten H., A. Golombek, A. Weigert, Franziska A. Franke, W. Westheide, G. Purschke, C. Bleidorn, and Kenneth M. Halanych. 2015. The Evolution of Annelids Reveals Two Adaptive Routes to the Interstitial Realm. *Current Biology* 25(15):1993-1999. doi: 10.1016/j.cub.2015.06.007
- Struck, T. H., and K. M. Halanych. 2010. Origins of holopelagic Typhloscolecidae and Lopadorhynchidae within Phyllodocidae (Phyllodocida, Annelida). *Zoologica Scripta* 39(3):269-275. doi: 10.1111/j.1463-6409.2010.00418.x
- Støp-Bowitz, C. 1948. Polychaeta from the "Michael Sars" North Atlantic deep-sea expedition 1910. Bergen Museum.
- Støp-Bowitz, C. 1991. Some New or Rare Species of Pelagic Polychaetes. In: *Systematics, Biology and Morphology of World Polychaeta: Proceedings of the 2nd International Polychaete Conference, Copenhagen, 1986*. p 261.
- Takagi, D. 2015. Swimming with stiff legs at low Reynolds number. *Physical Review E* 92(2):023020. doi: 10.1103/PhysRevE.92.023020
- Tzetlin, A. B., T. Dahlgren, and G. Purschke. 2002. Ultrastructure of the Body Wall, Body Cavity, Nephridia and Spermatozoa in Four Species of the Chrysopetalidae (Annelida, "Polychaeta"). *Zoologischer Anzeiger - A Journal of Comparative Zoology* 241(1):37-55. doi: 10.1078/0044-5231-00018
- Tzetlin, A. B., and A. V. Filippova. 2005. Muscular system in polychaetes (Annelida). *Hydrobiologia* 535(1):113-126. doi: 10.1007/s10750-004-1409-x
- Weidhase, M., C. Bleidorn, P. Beckers, and C. Helm. 2016. Myoanatomy and anterior muscle regeneration of the fireworm *Eurythoe* cf. *complanata* (Annelida: Amphinomidae). *Journal of Morphology* 277(3):306-315. doi: 10.1002/jmor.20496
- Weigert, A., A. Golombek, M. Gerth, F. Schwarz, T. H. Struck, and C. Bleidorn. 2016. Evolution of mitochondrial gene order in Annelida. *Molecular Phylogenetics and Evolution* 94:196-206. doi: 10.1016/j.ympev.2015.08.008
- Westheide, W. 1997. The direction of evolution within the Polychaeta. *Journal of Natural History* 31(1):1-15. doi: 10.1080/00222939700770011
- Yang, R. 2012. Fluid dynamic research on polychaete worm, *Nereis diversicolor* and its biomimetic applications, University of Bath.
- Zhang, Y., J. Sun, C. Chen, H. K. Watanabe, D. Feng, Y. Zhang, J. M. Y. Chiu, P.-Y. Qian, and J.-W. Qiu. 2017. Adaptation and evolution of deep-sea scale worms (Annelida: Polynoidae): insights from transcriptome comparison with a shallow-water species. *Scientific Reports* 7(1):46205. doi: 10.1038/srep46205
- Zhang, Y., J. Sun, G. W. Rouse, H. Wiklund, F. Pleijel, H. K. Watanabe, C. Chen, P.-Y. Qian, and J.-W.

Qiu. 2018. Phylogeny,
evolution and
mitochondrial gene
order rearrangement
in scale worms
(Aphroditiformia,
Annelida). *Molecular*
*Phylogenetics and*
*Evolution* 125:220-
231. doi:
10.1016/j.ympev.2018
14 .04.002

R. Soc. open sci. article template

Figure and table captions

Figure 1. Schematic representation of the examined parapodial myoanatomy. Inspired by the results of *Harmothoe imbricata* (Barnich et al., 2006). All muscle groups are colour coded and numbered according to the legends. Dorsal is up in both drawings. (A) External parapodial myoanatomy. (B) Internal parapodial myoanatomy. Posterior bundles for simplicity are not shaded darker in this schematic illustration. Abbreviations; ac, acicula; dlm, dorsal longitudinal muscle; vlm, ventral longitudinal muscle.

Figure 2. External and internal myoanatomy of the middle segment, right side parapodium in *Gesiella jameensis*. Volume rendering based on microCT data in (A) lateral view, (B, D) anterior view, (C, E) posterior view. Dorsal is up in all images. Body surface is semi-transparent. (A) Overview of complete parapodial myoanatomy with volume rendering of preceding and succeeding parapodia. (B, C) External parapodial myoanatomy. (D, E) Internal parapodial myoanatomy. Color coding and numbering; #I, anterior dorsal muscle, blue; #II, posterior dorsal muscle, dark blue; #III, anterior ventral muscle, green; #IV, posterior ventral muscle, dark green; #V-VI, oblique muscles, blue-green; #VII, preceding and succeeding bracing muscles, purple; #VIII, additional bracing muscles, purple; #IX, dorso-ventral crossing muscle, dark red; #X, anterior-posterior transverse muscles, turquoise; #XI, ventral diagonal muscles, lime green; #XII, notopodial acicula muscles, yellow; #XIII, notopodial protractor muscles, pink; #XIV, notopodial retractor muscle, brown; #XV, notopodial chaetal sac muscle, beige; #XVI, neuropodial acicula muscles, orange; #XVII, neuropodial protractor muscles, pink; #XVIII, notopodial retractor muscle, brown; #XIX, neuropodial chaetal sac muscle, beige; #XX, elyrophore muscles, olive green, (depressor muscle, egg white colored); #XXI, interacicular muscle, white. Abbreviations; ac, acicula; dlm, dorsal longitudinal muscle; s, suspensor muscles; vlm, ventral longitudinal muscle. Scale bars: 500µm.

[revised manuscript text omitted]

Table 3. Parapodium morphometric overview. The table shows total muscle volume in mm³ (left) and the size correlated volume (volume relative to body width, corr.) (right), number of individual muscle bundles (left) and muscle groups (right) and relative parapodia length (parapodia length: body width ratio) for *Gesiella jameensis*, *Pelagomacellicephala iliffei*, *Macellicephala longipalpa*, *Harmothoe imbricata* and *Branchipolynoe* sp. P-value < 0.001 for relative parapodia length between species. * p-value < 0.05.

Table 4. Parapodial muscle volume overview. The table shows muscle volumes (mm³), the size correlated volume, which is measured as volume relative to body width, (corr.) and relative volume (%) for each muscle group of *Gesiella jameensis*, *Pelagomacellicephala iliffei*, *Macellicephala longipalpa*, *Harmothoe imbricata* and *Branchipolynoe* sp. For *Branchipolynoe* sp., volume of bracing muscle #VIII was merged respectively with muscle #VII because the muscle groups could not be distinguished from each other during reconstruction. Muscle groups and respective numbers are shown to the left.

Figure 1. Schematic representation of the examined parapodial myoanatomy. Inspired by the results of *Harmothoe imbricata* (Barnich et al., 2006). All muscle groups are colour coded and numbered according to the legends. Dorsal is up in both drawings. (A) External parapodial myoanatomy. (B) Internal parapodial myoanatomy. Posterior bundles for simplicity are not shaded darker in this schematic illustration. Abbreviations; ac, acicula; dlm, dorsal longitudinal muscle; vlm, ventral longitudinal muscle.

1579x856mm (96 x 96 DPI)

Figure 2. External and internal myoanatomy of the middle segment, right side parapodium in *Gesiella jameensis*. Volume rendering based on microCT data in (A) lateral view, (B, D) anterior view, (C, E) posterior view. Dorsal is up in all images. Body surface is semi-transparent. (A) Overview of complete parapodial myoanatomy with volume rendering of preceding and succeeding parapodia. (B, C) External parapodial myoanatomy. (D, E) Internal parapodial myoanatomy. Color coding and numbering; #I, anterior dorsal muscle, blue; #II, posterior dorsal muscle, dark blue; #III, anterior ventral muscle, green; #IV, posterior ventral muscle, dark green; #V-VI, oblique muscles, blue-green; #VII, preceding and succeeding bracing muscles, purple; #VII, additional bracing muscles, purple; #IX, dorso-ventral crossing muscle, dark red; #X, anterior-posterior transverse muscles, turquoise; #XI, ventral diagonal muscles, lime green; #XII, notopodial acicula muscles, yellow; #XIII, notopodial protractor muscles, pink; #XIV, notopodial retractor muscle, brown; #XV, notopodial chaetal sac muscle, beige; #XVI, neuropodial acicula muscles, orange; #XVII, neuropodial protractor muscles, pink; #XVIII, notopodial retractor muscle, brown; #XIX, neuropodial chaetal sac muscle, beige; #XX, elytophore muscles, olive green, (depressor muscle, egg white colored); #XXI, interacicular muscle, white. Abbreviations; ac, acicula; dlm, dorsal longitudinal muscle; s, suspensor

muscles; vlm, ventral longitudinal muscle. Scale bars: 500 μ m.

540x670mm (96 x 96 DPI)

50 531x671mm (96 x 96 DPI)

Figure 4. Comparison of muscle groups #VII-XI for *Gesiella jameensis*, *Pelagomacellicephala iliffei*, *Macellicephala longipalpa*, *Harmothoe imbricata* and *Branchipolynoe* sp. Volume rendering based on microCT data from mid-parapodium with aciculae (pipe structure, black). Dorsal is up in all images except for muscle #XI, which is shown in ventral view. Muscle group #VIII is not included in *Branchipolynoe* sp. since it is merged with muscle #VII. Numbers 1-11 are for individual muscle bundles. Colour coding and numbering; #VII, preceding and succeeding bracing muscles, purple; #VII, additional bracing muscles, purple; #IX, dorso-ventral crossing muscle, dark red; #X, anterior-posterior transverse muscles, turquoise; #XI, ventral diagonal muscles, lime green.

541x671mm (96 x 96 DPI)

Figure 5. Comparison of muscle groups #XII-XX for *Gesiella jameensis*, *Pelagomacellicephalo iliffei*, *Macellicephalo longipalpa*, *Harmothoe imbricata* and *Branchipolynoe* sp. Volume rendering based on microCT data from mid-parapodium with aciculae (pipe structure, black). Dorsal is up in all images. Numbers 1-10 are for individual muscle bundles. Color coding and numbering; #XII, notopodial acicula muscles, yellow; #XIII, notopodial protractor muscles, pink; #XIV, notopodial retractor muscle, brown; #XV, notopodial chaetal sac muscle, beige; #XVI, neuropodial acicula muscles, orange; #XVII, neuropodial protractor muscles, pink; #XVIII, notopodial retractor muscle, brown; #XIX, neuropodial chaetal sac muscle, beige; #XX, elytophore muscles, olive green, (depressor muscle, egg white coloured). Abbreviations; s, suspensor muscles.

540x673mm (96 x 96 DPI)

Figure 6. Comparison of the interacicular muscle #XXI for *Harmothoe imbricata* and *Branchipolynoe sp.* Volume rendering based on microCT data from mid-parapodium with aciculae (pipe structure, black). Dorsal is up in all images and shown from proximal view. Muscle #XXI is lacking in *Gesiella jameensis* and *Pelagomacellicephala iliffei*. Colour coding and numbering; #XXI, interacicular muscle, white. Abbreviations; ac, acicula.

342x155mm (96 x 96 DPI)

Figure 7. Locomotion of *Gesiella jameensis*, *Pelagomacellicephala iliffei* and *Harmothoe imbricata* in lab setup. (A) shows video frames of swimming *P. iliffei* (left), *G. jameensis* (middle) and *H. imbricata* (right) in dorsal view. (A, B). Numbers 1-4 refer to parapodia movement during swimming (see results, section 4.3). (B) Showing the video frame of swimming *P. iliffei* in lateral view. (C) Frame shot of crawling *H. imbricata* in lateral view. Numbers 1-5 refer to parapodial movements during crawling steps (see results, section 4.3).

166x149mm (220 x 220 DPI)

Figure 8. In situ frame shots from *Gesiella jameensis* within Cueva de los Lagos, Lanzarote, Canary Islands, Spain. (A) *Gesiella jameensis* motionless with spread dorsal cirri and downward hanging parapodia. (B) Swimming behaviour of *G. jameensis* after being disturbed by divers.

169x67mm (220 x 220 DPI)

Table 1.

Species	Sample locality	Year	Depth(m)	USNM	μCT	VREC	CLSM
Pelagaomacellicephalo iliffei	Grotto Hole, Long Island, Bahamas	2007	1-2	-	2	-	-
Pelagomacellicephalo iliffei	Cottage Pond, North Caicos, Turks and Caicos	2019	1.5	-	-	2	2
Gesiella jameensis	Túnel de la Atlántida, Lanzarote, Canary Island, Spain	2014	2-20	-	2	-	-
Gesiella jameensis	Cueva de los Lagos, Lanzarote, Canary Islands, Spain	2019	1.5	-	-	4	2
Macellicephalo longipalpa	70° 51' N. 52° 01' W, West Greenland	1928	733	51968	1	-	-
Harmothoe imbricata	Kaldbak, Kaldbak Fjord, Faroe Island	2018	-	-	2	3	3
Branchipolynoe sp.	37° 40' 21.0" S 110° 52' 37.2" W, South Pacific	2005	+2236	1463999	2	-	-

[revised manuscript text omitted]

* *p*-value < 0.05

Table 4. Parapodial muscle volumes

Muscle group number (#)	Muscle group	Gesiella jameensis			Pelagomacellicephala iliffei			Macellicephala longipalpa			Harmothoe imbricata			Branchipolynoe sp.		
		Vol. (mm ³)	Vol. (corr.)	Relative vol. (%)	Vol. (mm ³)	Vol. (corr.)	Relative vol. (%)	Vol. (mm ³)	Vol. (corr.)	Relative vol. (%)	Vol. (mm ³)	Vol. (corr.)	Relative vol. (%)	Vol. (mm ³)	Vol. (corr.)	Relative vol. (%)
#I	Anterior dorsal muscle	0.0061	0.0081	13.8	0.0048	0.0059	13.7	0.1892	0.0943	15.7	0.0136	0.0100	9.0	0.1105	0.0460	15.3
#II	Posterior dorsal muscle	0.0060	0.0081	13.8	0.0047	0.0059	13.6	0.1497	0.0746	12.4	0.0138	0.0101	9.1	0.0925	0.0385	12.8
#III	Anterior ventral muscle	0.0079	0.0105	17.9	0.0054	0.0068	15.6	0.1696	0.0846	14.1	0.0125	0.0092	8.3	0.0732	0.0305	10.1
#IV	Posterior ventral muscle	0.0062	0.0083	14.2	0.0042	0.0053	12.2	0.1641	0.0818	13.6	0.0092	0.0068	6.1	0.0507	0.0211	7.0
#V	Anterior oblique muscle	0.0008	0.0010	1.8	0.0012	0.0015	3.4	0.0124	0.0062	1.0	0.0135	0.0099	8.9	0.0261	0.0109	3.6
#VI	Posterior oblique muscle	0.0007	0.0009	1.5	0.0012	0.0015	3.4	0.0149	0.0074	1.2	0.0125	0.0092	8.2	0.0246	0.0102	3.4
#VII	Preceding and succeeding bracing muscles	0.0068	0.0091	15.5	0.0041	0.0051	11.7	0.1924	0.0959	16.0	0.0063	0.0047	4.2	0.1849	0.0770	25.6
#VIII	Additional bracing muscle	0.0015	0.0020	3.3	0.0010	0.0012	2.8	0.0332	0.0166	2.8	0.0026	0.0019	1.7	N/A	N/A	N/A
#IX	Dorso-ventral crossing muscle	0.0003	0.0005	0.8	0.0004	0.0005	1.2	0.0080	0.0040	0.7	0.0065	0.0048	4.3	0.0303	0.0126	4.2
#X	Anterior-posterior transverse muscles	0.0009	0.0012	2.1	0.0008	0.0010	2.3	0.0292	0.0146	2.4	0.0047	0.0035	3.1	0.0195	0.0081	2.7
#XI	Ventral diagonal muscles	0.0006	0.0008	1.3	0.0008	0.0010	2.3	0.0268	0.0133	2.2	0.0048	0.0035	3.2	0.0222	0.0092	3.1
#XII	Notopodial acicula muscles	0.0013	0.0018	3.1	0.0011	0.0014	3.1	0.0301	0.0150	2.5	0.0099	0.0073	6.5	0.0168	0.0070	2.3
#XIII	Notopodial chaetal protractor muscles	0.0003	0.0003	0.6	0.0000	0.0000	0.1	0.0014	0.0007	0.1	0.0006	0.0004	0.4	0.0002	0.0001	0.0
#XIV	Notopodial chaetal retractor muscles	0.0001	0.0001	0.2	0.0011	0.0014	3.2	0.0018	0.0009	0.2	0.0035	0.0026	2.3	0.0043	0.0018	0.6
#XV	Notopodial chaetal sac muscles	0.0004	0.0006	1.0	0.0001	0.0001	0.3	0.0154	0.0077	1.3	0.0023	0.0017	1.5	0.0047	0.0020	0.7
#XVI	Neuropodial acicula muscles	0.0022	0.0029	5.0	0.0012	0.0015	3.4	0.0440	0.0220	3.7	0.0096	0.0071	6.4	0.0213	0.0089	3.0
#XVII	Neuropodial chaetal protractors muscles	0.0002	0.0003	0.5	0.0004	0.0005	1.2	0.0145	0.0072	1.2	0.0041	0.0030	2.7	0.0051	0.0021	0.7
#XVIII	Neuropodial chaetal retractor muscles	0.0007	0.0010	1.6	0.0006	0.0007	1.7	0.0163	0.0081	1.4	0.0043	0.0032	2.8	0.0083	0.0034	1.1
#XIX	Neuropodial chaetal sac muscles	0.0005	0.0006	1.1	0.0010	0.0013	2.9	0.0613	0.0306	5.1	0.0065	0.0048	4.3	0.0071	0.0030	1.0
#XX	Elytrophore muscles	0.0004	0.0005	0.9	0.0007	0.0009	2.0	0.0282	0.0141	2.3	0.0099	0.0073	6.5	0.0188	0.0079	2.6
#XXI	Interacicular muscle	N/A	N/A	N/A	N/A	N/A	N/A	N/A	N/A	N/A	0.0005	0.0004	0.3	0.0003	0.0001	0.0
	all muscles	0.0439	0.0585	100.0	0.0347	0.0434	100.0	1.2026	0.5995	100.0	0.1511	0.1111	100.0	0.7215	0.3006	100.0

Appendix B

Rebuttal letter (point by point response) to reviews of paper RSOS-210541

Dear editor and reviewers. Thank you for your thoughtful and constructive criticism and suggestions, which we have nearly all followed as stated below.

Associate Editor Comments to Author (Dr Jonas Rubenson):

Comments to the Author:

Dear Dr. Worsaae,

As you will see, the reviewers are overall positive about your paper and all three reviewers find your experiment interesting. However, the reviewers raise some concerns that you will need to address upon revision. I am sympathetic to the reviewer's concern about the clarity of hypotheses. The suggestions to include a description of both power stroke and return stroke, and a description of the parapodial muscle complex are also useful.

Hopefully the reviewers' comments, which I believe are constructive, are addressable upon revision.

Reviewer comments to Author:

Reviewer: 1

Comments to the Author(s)

This is an interesting study of the potential morphological changes associated with swimming in annelid worms. The authors studied the morphology and movements of five scale worm species that show a range of locomotion including the typical benthic crawling of annelids, occasional swimming, and continuous swimming. The morphology of the musculature of the parapodia was investigated with confocal laser scanning microscopy and microCT. The microCT data were used in 3D reconstruction and in estimates of muscle volume. Video recordings of swimming were made in several of the species for which live animals were available. The morphology of the musculature was described, along with estimates of the volume of the various muscles. Basic aspects of the swimming movements were described from the video recordings. The results were discussed in the context of attempting to identify specializations associated with swimming in annelids.

The study provides a detailed description of the numerous (21) muscles of the parapodia in the five species, taking good advantage of the remarkable morphological data provided by microCT methods. The 3D reconstructions of the muscles included in the figures are nicely presented, help in understanding the rather complicated morphology, and represent a significant contribution to our understanding of annelid muscular morphology. Although the individual panels in the figures are a bit small, since the majority of readers will view the figures online and can adjust the size on screen, I do not believe that the size will be a problem, especially since readers can consult the more extensive morphological descriptions and excellent figures in the supplemental materials.

The manuscript overall is quite well written and clear. The interpretation of the role of the various muscles in locomotion is reasonable, but it is necessarily limited by the fact that

most of the techniques (electromyography, sonomicrometry, muscle mechanics, x-ray imaging, strain gauge measurement, etc.) are not feasible on these relatively small specimens. I have included a few minor comments and suggestions below.

Page Line Comment

2 38 Revise to read: "... but are negatively buoyant and thus cannot passively maintain.."

Thank you, corrected

5 30 It would be helpful to include the internal diameter of the pipette tip here to provide some sense of the volume scanned.

Thank you, corrected

I suspect that many readers will be like me and not be familiar with the size of the species described in the study. In general it would be helpful to add scale bars to the 3-D reconstruction figures and the live animal images.

We agree. Scale bars were already added to the parapodial overview 3-D reconstructions in Fig. 2 (now a revised version) and Suppl. Figs 1-6, to give the reader a better understanding of the parapodial size and the individual muscles. We therefore don't find it necessary to add scale bars to the individually illustrated muscles in Figs 3-6. However, we have now added specimen sizes to Table 1 in the method section as well as added estimated size bars to the live images in Figs 7-8.

10 37 As is mentioned later in the paper, the reduction in drag of the parapodia on the return stroke is critical to providing a net thrust for swimming. It would thus be of interest to provide a more detailed description here on your observations of the folding and unfolding of the chaetae during the return and power strokes.

Agree, added brief description in section 4.3 on the position of the chaetae during power and recovery strokes and a brief discussion in section 5.2.

Did you note differences between the species studied?

The behavior of chaetae during swimming were only possible to detect for specimens of *Pelagomacellicephala iliffei*.

This issue of the relative drag of the two strokes is particularly important given the small size, and thus relatively low Reynolds Number that these animals experience. It would also be of interest to note any differences in the relative duration of the power and return strokes. At high Reynolds numbers a slower return stroke relative to the power stroke can provide net thrust, but this works less and less well as Reynolds numbers decrease; at low Reynolds numbers reconfiguration to reduce drag on the return stroke is essential.

Although all of the studied species are muscular macrofaunal species, this is a very interesting point and we have briefly discussed the subject in section 5.2. However more lab data would be needed for detailed quantitative analysis, since it is difficult to obtain suitable, in-focus lateral imagery of the animals swimming freely and of the water currents generated by them. This also falls outside the scope of this paper, but would be interesting for further study.

14 20 Revise to read: "...support crawling and aid in twisting, bending, and constriction..."

Ok, done

14 52,54 Replace "less' with "fewer"

OK, done

15 8 It might be worthwhile to consider the implications of the elongation of the muscles for the output of the muscles in swimming. Because of the implications of series vs parallel elements in muscle, the output of a muscle depends greatly on its dimensions. For example, compare two muscles that are identical in fiber type and in volume but muscle "A" is twice as long as muscle "B". The shortening velocity of muscle A will be twice that of muscle B. But muscle B will have a greater cross-sectional area (because we are holding volume constant in this case) and thus its peak isometric tension will be twice that of muscle A. Interestingly, since power is force times velocity the two muscles will have the same power output. This is also why I made the comment above about providing additional detail on the velocity of the power and return strokes. For instance, are the power strokes of the parapodia of the swimming animals faster than those of the benthic crawlers? The elongated muscles of the swimming animals would facilitate this. The stouter muscles of the crawlers would be capable of greater force production, assuming that they do indeed have a larger relative cross-sectional area.

Again, this a very interesting comment and comments on swimming speed /parapodial speed is briefly added to both section 4.3 and 5.2 since the swimming scale worms do seem to have much faster swimming speed and parapodial movements. However, we would need more quantitative data for accessing the velocity quantitatively and unfortunately a detailed comparison of parapodial velocity and swimming speed is not possible with the ranging quality of our recordings. Since these animals are generally difficult to collect and image behaviour of at adequate resolution, the anatomical reconstructions rather than quantification of the muscle movements were the focus of this paper.

Reviewer: 2

Comments to the Author(s)

Dear authors, your manuscript represents a well-written, detailed and highly needed comparative investigation of an understudied character complex in annelids. The data is well-presented and illustrated, results are carefully described and the conclusions are reasonable and embedded in recent knowledge. I therefore recommend publication of this nice contribution - there are just few minor points that should be addressed before final acceptance (see attached).

Ok, the suggestions in the attachment is all followed

In my opinion the parapodial musculature has to be discussed together with the body wall musculature as well - at least some words should mention the overall arrangement of the main body-wall muscles around the parapodium and how the parapodial muscle complex is embedded in this arrangement. This would help the reader to get a better orientation and can possibly help to compare the entire structure with other taxa in future studies.

Ok, body wall musculature is further mentioned in the introduction alongside description of parapodial movements. The body wall muscles are addressed briefly in result and discussion section

Reviewer: 3

Comments to the Author(s)

This article provides an interesting and novel look at the adaptations required for polynoid annelids to develop a continuously swimming lifestyle. The authors convincingly show that the parapodial musculature of continuous swimmers differs from those of crawlers or burrowers in ways that make sense for these different lifestyles, and that differences occur in the relative size of muscles rather than through changes in the types of muscles or in locomotion in the individuals examined. The manuscript contains a lot of really interesting data, and my comments are primarily focused on clarifying what is, on first read, an overwhelming amount of information, to clearly convey the really interesting findings of this paper.

(note that it's a bit confusing because the author page numbers are different from the pdf page numbers and the line numbers don't match the text – I'm using the author page numbers and approximate line numbers)

1. The introduction lacks a clear hypothesis. The goals currently presented (page 4 lines 22-27) are overly general and do not explicitly state the theory being tested, making it difficult to understand the reasoning behind the myoanatomical comparisons made in the results and discussion. From reading the paper, it seems like there are two clear hypotheses that are not necessarily mutually exclusive and are both very interesting – (1) the transition from crawling to swimming involves changes in locomotory behavior, specifically of the parapodia, and the underlying musculature versus (2) that the transition to swimming involves dealing with the problem of buoyancy, in which case the (relatively dense) muscle volume should be reduced in swimmers. The findings primarily support #2 but with some interesting differences in specific muscles consistent with #1.

We have now rephrased parts of the introduction and aims to more clearly present the overall hypotheses.

2. The Figure 1 schematic is really helpful, and would be stronger if it were integrated into more explicit hypotheses, e.g., which muscles would one expect to be larger or smaller in swimmers versus crawlers?

We have in our predictions refrained from being too specific in regards to individual muscle groups since the literature really did not provide any basis for such prediction. In fact, these more specific hypotheses are the ones we seek to establish through our detailed comparative study, and which we now more clearly summarise in our conclusion. However, according to your advice we have introduced a comparative Fig.2, which indicates what to expect from the following more detailed comparisons.

There's a lot of information in this manuscript, and while the muscle groups are complex and appropriately named, maybe they could be further categorized by predicted differences?

We understand the idea, and have partly followed it by now changing Fig. 2 to illustrating a parapodium of a swimming versus a crawling species. This figure hints at which muscles can be expected to differ between the two species. Moreover, we have on the Fig. 3-5 and across all species, marked selected common states (e.g., elongated/thin/dense) later hypothesised to be major adaptations to swimming or crawling.

I do really like the consistent use of color across the figures to keep track of which muscles are which.

Thank you 😊

3. Comparison of species within a family needs to be put in a phylogenetic context. For example, are the two swimming species closely related, or did swimming evolve more than once within the family? A figure showing the phylogeny of the examined worms (and relevant traits) would be extremely useful.

We have actually been in doubt on how much to discuss ancestry in this study, but generally found it misleading to enter a tree of the groups since so many other polynoid groups would be missing from such a tree and the internal relationships are still highly debated (see e.g. Gonzalez et al. 2021, now in ref list). However, we did actually inform on the known relationships through giving subfamily names and have now inserted a “sister” where forgotten in the introduction. Moreover, we now have inserted several references to their heritage throughout the ms – e.g. in 5.1., 5.2 and in the conclusion.

4. While the methods section was technically sufficient, it lacked justification and it was not clear how each section addressed the various goals of the research. For example, even after reading the results and discussion, it was not apparent how the confocal laser scanning microscopy contributed to this research. This could be easily addressed after the addition of hypotheses, by linking each of the various methods to the individual hypothesis that were tested. I suggest adding a sentence at the beginning of each paragraph to give a brief overview of why the methods were used before getting into the technical details.

I suggest adding a sentence at the beginning of each paragraph to give a brief overview of why the methods were used before getting into the technical details.

A brief descriptions of the used methods is added in the end of the introduction and in the beginning of each paragraph.

The video analysis section focuses on the video recording methods and does not actually discuss any analyses of videos other the the last sentence which mentions the software but not what was actually measured or characterized.

Thanks for pointing this out, further explanation has been added to the last paragraph in section 3.5

5. When calculating the size correlated measure of muscle volume (page 6 lines 25-27), the authors divided muscle volume by segment width, resulting in a measure of area which is difficult to interpret and does not allow for comparison across different sized species. It would be more appropriate to present a non-dimensional measure of volume by dividing muscle volume by the volume of the entire corresponding segment (or some proxy of volume, e.g., length x width x thickness). This non-dimensional metric of muscle volume could then directly test the hypotheses.

We agree, and have now made a new Corr. Measure that agrees to your proposal. We have divided the muscles volumes with the respective segment volume, which shows an easier non-dimensional metric.

6. The statistics are somewhat confusing. I am uncertain what measurements are being compared in the t-test mentioned on page 6 lines 27-32. The results of this test are not mentioned anywhere other than in table 3 caption, and a more thorough explanation should be added.

A more thorough explanation of the tests is now added to end of section 3.4 and a brief description of the result of the tests of the parapodia length is added to section 4.2

Table 3 gives p-values, but for what comparison?

Now explained

7. Figures 3-6 show the differences in muscle bundle size for each of the species examined, however the figures are a bit overwhelming because the comparisons / conclusions the reader should draw are not immediately apparent. These figures would be greatly improved if they were either constrained to only the muscle bundles that illustrate the differences between a swimming and crawling lifestyle, or edited to draw more attention to the relevant comparisons. Perhaps consider moving the full figures to the supplement and making one figure that highlights the most interesting comparisons to include in the main text?

We agree on clarifying and have now made a new comparative Fig.2 (and moved the original fig. 2 to supplementary material) and on all the following figs added a text indicating whether the species shown is a swimming or crawling species. Moreover, we have on figs 3-5 added labels on the figs indicating the hypothesized adaptations.

It would be helpful to indicate with the species names which are crawlers and which are swimmers so the reader doesn't have to flip back to earlier sections to recall.

Done, see fig. 3-6

Alternately, focus more on two species, one crawler and one swimmer, and put much of the data from the other 3 species in the supplement to simplify the message.

Bold idea, and although we follow your line of thought, we believe the paper is more about establishing hypotheses based on comparative evidence - rather than testing a specific hypotheses. Therefore, it is important to illustrate that we compare more than just two species and we believe simplifying the ms to this extend would actually devaluate the results and conclusions. We also believe that not everything in Science has to be explained on two pages.

8. I found that comparing Figure 2 to the supplementary figure S3 really clearly illustrated the difference in muscle volume between the swimming and crawling worms. I suggest moving one of the supplementary figures into the main body of the text to allow for comparison of musculature between swimmers and crawlers.

Thank you for the good idea, we have now added a new figure 2 that compares the old fig. with S3 directly in order to highlight the differences and make comparison easier

9. I found table 4 to be too large and cumbersome to be useful. Multiple columns for each species make comparisons very difficult. A reduced table focusing on a single characteristic (I suggest normalized dimensionless muscle volume), or better yet a figure summarizing the salient points of the table would be much more useful. Table 4 should be moved to the supplement.

We moved table 4 to the suppl. Mat. and therefore dont find it necessary to reduce the given measurements, however the corr. Has now been made a dimensionless measure (muscle vol / segment vol).

10. Table 2 is also fairly large and cumbersome – consider moving it to the supplement and presenting a simpler summary table or figure here. For example, consider modifying the Fig. 1 schematic to show the key differences between crawlers and swimmers, potentially focusing on one representative crawling and one swimming species.

As argued for above, we find it misleading to present detailed hypotheses on muscular functions on Fig.1 when it is actually the aim of this paper to achieve a comparative understanding (summarized in our figs and Table 2) that can help establish such hypotheses. Yet, we have partly followed this idea by modifying Fig. 2 according to your suggestions.

11. Finally, the discussion makes a lot of good points, but quite a bit of the material in the discussion could be moved to the introduction in laying out clear, specific hypotheses, allowing for more elaboration on the broader implications of the work in the discussion. For example, while the reduction in muscle bundles and density shown is really interesting, how do those effects scale across the entire worm? Do these reductions reduce overall the density of the worm by an amount relevant to swimming vs crawling?

We have added a brief comment on repeated segment features resulting in overall density loss in the discussion

12. The title could be edited to more clearly convey the main point and reduce jargon (remove “myoanatomy”), e.g., “Reduced parapodial muscle volume facilitates swimming in Polynoidae”

Title changed

Kelly Dorgan